# New Records and Updated Checklist of the Pentatomoidea (Hemiptera: Heteroptera) of Greece

**DOI:** 10.3390/insects13080749

**Published:** 2022-08-19

**Authors:** Antonios Tsagkarakis, Zoi Thanou, Aikaterini Chaldeou, Ioanna Moschou, Argyro Kalaitzaki, Sakis Drosopoulos

**Affiliations:** 1Laboratory of Agricultural Zoology and Entomology, Agricultural University of Athens, Athens 118 55, Greece; 2Laboratory of Sericulture and Apiculture, Agricultural University of Athens, Athens 118 55, Greece; 3Laboratory of Agricultural Entomology, Institute for Olive Tree, Subtropical Plants and Viticulture, Hellenic Agricultural Organization ‘DEMETER’, Agrokipio, Chania 731 00, Greece; 4Laboratory of Genetics, Agricultural University of Athens, Athens 118 55, Greece

**Keywords:** Acanthosomatidae, Cydnidae, Pentatomidae, Plataspidae, Scutelleridae, burrower bugs, jewel bugs, shield bugs, stink bugs, new records, Greece

## Abstract

**Simple Summary:**

An updated species checklist for all the Pentatomoidea species for Greece is provided. Eight species are recorded from Greece for the first time. The checklist is supported with distributional data notes for all the Pentatomoidea species of Greece.

**Abstract:**

Eight species of the superfamily Pentatomoidea are recorded from Greece for the first time: *Aelia germari* Küster 1852, *Eurygaster hottentotta* (Fabricius 1775), *Eysarcoris aeneus* (Scopoli 1763), *Neottiglossa lineolata* (Herrich-Schaeffer 1830), *Neottiglossa pusilla* (Gallen 1789), *Picromerus bidens* (Linnaeus 1758), *Podops (Podops) inunctus* (Fabricius 1775) and *Tarisa pallescens* (Jakovlev 1871). A complete updated species checklist with distributional data notes for all the new species for Greece are provided.

## 1. Introduction

The superfamily Pentatomoidea constitutes one of the most important insect groups of the suborder Heteroptera. It includes 1080 genera and 5907 species belonging to 16 families of which the Cydnidae, Pentatomidae, Scutelleridae and Tessaratomidae are the most important; 94% of the species belong to these four families [1,2,3]. Pentatomidae is the fourth most numerous family within Heteroptera with more than 4700 species, commonly known as stink bugs [4]. It includes predominantly herbivorous species, some of them having high economic importance, such as the brown marmorated stink bug *Halyomorpha halys* (Stål) and the southern green stink bug *Nezara viridula* (L.).

*Halyomorpha halys* is a polyphagous stink bug native to China, Korea, Japan and Taiwan [5]. It was introduced into the United States in the mid-1990s, resulting in losses of apples, peaches, tomatoes and other important crops [6]. Recently, it has been detected in Europe for the first time in Switzerland in 2007, damaging stone fruits and legume pods [7]. Until recently, it has been recorded in many European countries including Greece [8], and it is considered a potential threat to agricultural productivity in all these regions [9,10] since it has a broad host range and is capable to disperse long-distance by flight [11].

There have been sporadic references on specific species of Pentatomoidea in Greece [8,12,13,14,15,16,17,18,19,20,21,22,23,24,25,26,27,28,29,30,31,32,33,34,35,36,37,38,39,40,41,42,43], and two overall studies, one in 1980 [44] and the other in 2019, which updated the previous one and included all new species [45]. Given the worldwide problem of the brown marmorated stink bug pest status [6] and its presence in Greece [8], understanding this family in Greece may assist in distinguishing this pest species from native ones, and provide more targeted response for future detections of this pest. The aim of the present study is to present new records and an updated checklist of the Pentatomoidea of Greece.

## 2. Materials and Methods

Insect material was collected using sweeping nets and glass containers, or by glass tube aspirators during 1974–2015. Insects were killed in glass tubes with ethyl-acetate or were directly stored in 70% ethyl-alcohol. Genitalia were prepared for observation under a Carl Zeiss Stemi 305 binocular stereoscope and Olympus CX23 binocular microscope by maceration in 10% potassium-hydroxide (KOH). All specimens were identified by the authors, according to keys and descriptions in references [46,47,48,49,50].

Material of the collected species was deposited in the collection established by the fourth author, which is kept in the Laboratory of Agricultural Zoology and Entomology of the Agricultural University of Athens.


*Localities studied*


Collection sites for each species are presented in the same fashion as Drosopoulos et al. [50]. The altitudes and distances reported for the collection sites may deviate ca. 50 m and ca. 500 m respectively.

The localities from which specimens were collected are the following:**NORTHWESTERN GREECE****Kerkyra:** Tritsi; Liapades**Ioannina:** Voutsaras (500 m); Pindos (Katara); Miliotades (600 m); Vrysochori (1000–1600 m); Aristi; Aoos-ΝΕ Konitsa; Vikos gorge (E Monodendri); Ioannina; Millies; Vouchorina; Korydallos; Votonossi**Florina:** Ε Pisoderion (1650 m); Kalo-Nero (Vernon Mt; southern slopes; 1100 m); Florina; Megali Prespa (Psarades); Ladopotamos valley (Ν Kotas); Vigla; Pisoderion; Vevi; Varnous Mt. (Bela Voda)**Kastoria:** Gavros (Ladopotamos valley); Aposkepos (800 m)**Kozani:** 15 km SE Siatista; Vourinos Μt. (1000–1300 m)**Grevena:** Anixis; Agii Theodori**Trikala:** Orthovounion; Aghiofilo; Mourgani**Larisa:** Olympus Μt. (southwestern slopes; Kryovryssi; 800 m)**Pieria:** Olympus Μt. (eastern slopes; Stavros); Olympus Μt. (eastern slopes; Prionia; 1000–1250 m); Poroi (coastal swamp); Litochoro**Pella:** Vryta; Ν Vegoritis Lake**WESTERN GREECE****Epirus:** Arta (Louros Riv; near Agios Georgios); Vryssochori**Aetoloakarnania:** Messolonghion; Messolonghion (Tourlida); Amphilochia (Anoixiatiko); Babini; Evinos Riv. (Evinochori); Astakos (Mytikas); Vonitsa (Loutrakion); Aetolikon; Trichonis (Kapsorachi)**Kephallinia:** Aenos Mt. (600–1000 m); K. Katelios; Myrtos; Assos**NORTHEASTERN GREECE****Evros:** Metaxades; Evros Delta**Samothraki:** SW Kamariotissa (coastal swamp); Loutra**Xanthi:** W Porto Lagos**Drama:** Rodopi Μt. (fountain “Krya Vryssi” near Elatia); Rodopi Μt. (13 km SE Elatia); Rodopi Μt. (Vathyrhemma; 1450 m); Falakron Mt. (2000–2200 m); Falakron Μt. (1400–1600 m); Rodopi Μt. (ΝΕ Elatia in direction Vathyrhemma; 1300 m); Rodopi Μt. (Betula Forest; 25 km SE Elatia; 1100 m); Rodopi Μt. (ΝΕ “Virgin Wood”; supra Zagrantenia; 1800–1900 m); S Livaderon 600 m; Ν Drama 450 m; Potami; NW Silli (Prasinada; 800 m)**Thessaloniki:** Cedron Hills; Plagiarion**CENTRAL GREECE****Attiki:** Votanikos; Alimos; Kifissia; Amaroussion; Chalandri; Avlon; Mati; Marathon; Marathon (coastal swamp and salt marshes near Schinias); Parnis Μt. (1200 m); Parnis Mt. (Mola fountain); Ymitos Mt. (Kareas)**Voeotia:** Arachova; E Agia Paraskevi; Aliartos; Tsoukalades; Avlis (Vathy)**Fokis:** Monastirakion (Doris; coastal swamp); Skaloula (Doris; 600 m); Eratini (Doris); Kalion (Doris); Eleon; Galaxidi; Amfissa; Agios Nikolaos (Doris); NW Itea (Agia Euthymia); Ghiona Mt. (near refuge); Parnassos Mt. (National Park); Marathias**Fthiotis:** Kalamakion; Oiti Mt. (1200–1600 m); Malessina; Kalamakion; Arkitsa**Karditsa:** Artessiano**PELOPONNESOS****Korinthia:** Killini Μt. (supra Trikala; 1550–1600 m); Akrokorinthos; Derveni**Achaia:** Bouboukas; Kato Klitoria**Llia:** Krestena; swamp near Loutra Kaiafa; Kyllini; SE Pyrgos (Paralia Zacharo)**Messinia:** NE Kalamata (Artemissia); E Kyparissia (Dorion); Kazarma; Menina**Arkadia:** Menalon Mt. (5 km W Vitina; 900–1000 m); SW Astros (Platanos); Tripolis; N Astros (Xeropigado); Parnon Μt. (Ν. Kosmas; 1100 m); Charadros; Parnon Μt. (supra Kastanitsa; 1100 m)**Lakonia:** Mystras; Taygetos Mt. (600–1000m)**EAST AEGEAN ISLANDS****Ikaria:** W Gyaliscari (coastal swamp); Ε Raches (600 m)**Samos:** Pythagorion; Neochorion (5 km E Koumeika); S Psili Ammos (coastal swamp); Pyrgos (10 km E Koumeika); N Agios Konstantinos (coastal swamp); Chora**Chios:** Chora; E Mastichochoria (Armolia); Central Mastichochoria (Agios Georgios)**Lesvos:** Sykaminea (Skala); Andissa; Kalloni; Petra**CYCLADES ISLANDS****Naxos:** SE Moutsouna, Apeiranthos**Paros:** swamp near Paroikia; Monastirion**Santorini:** Kamarion**Syros:** Poseidonia**Tinos:** Triantaros**DODECANESOS ISLANDS****Rhodos:** Kremasti (coastal biotopes); Dimilia; Plymiri; Emponas; Nea Afantou; Apolakkia; Salakos; Petaloudes valley; Kalathos; Archipolis; Malonas**Kastellorizo:** around village**CRETE****Heracleon:** Agios Vassilios**Rethymnon:** Myloi; Agioi Apostoloi; Oidi Mt. (600–1000 m)**Chania:** Chryssopigi

## 3. Results

Results of this study showed that 92 species were found in 52 genera, belonging in 10 subfamilies, within 5 families of Pentatomoidea: Acanthosomatidae (Acanthosomatinae), Cydnidae (Cydninae, Sehirinae), Pentatomidae (Asopinae, Pentatominae, Podopinae), Plataspidae (Plataspinae) and Scutelleridae (Eurygastrinae, Odontoscelinae, Odontotarsinae). Eight of these species are new records for Greece: 6 species of the subfamily Pentatominae (Pentatomidae) and 2 species of Podopinae (Pentatomidae). The updated checklist with the new records is referred below in the text and in Table 1.

**Checklist of species found in the present study** (with bold and asterisk the new species record for Greece)


**Family ACANTHOSOMATIDAE**



**Subfamily Acanthosomatinae**


Genus *Cyphostethus* Fieber 1860 (Figure 1)

*Cyphostethus tristriatus* (Fabricius 1860)

Doris [Agios Nikolaos (20.VII.1986), Skaloula (27.III.1979)], Olympos Mt. [Prionia (21.V.1981)]. Total: 3 specimens.

Genus *Elasmucha* Stål 1864 (Figure 1)

2.*Elasmucha grisea grisea* (Linnaeus 1758)

Doris [Skaloula (2.V.1986)], Ioannina (24.VI.1984), Rodopi Mt. (10.VIII.1985). Total: 5 specimens.


**Family CYDNIDAE**



**Subfamily Cydninae**


Genus *Cydnus* Fabricius 1803 (Figure 2)

3.*Cydnus aterrimus* (Forster 1771)

Santorini Island [Kamarion (5.V.1982)]. Total: 1 specimen.

Genus *Macroscytus* Fieber 1860 (Figure 2)

4.*Macroscytus brunneus* (Fabricius 1803)

Attiki [Kifissia (2.X.1981, 26.VIII.1985)], Doris [Agios Nikolaos (2.VII.1986)]. Total: 4 specimens.


**Subfamily Sehirinae**


Genus *Ochetostethus* Fieber 1860 (Figure 2)

5.*Ochetostethus balcanicus* (Wagner 1940)

Doris [Agioi Pantes (26.V.1980), Skaloula (29.IV.1978, 1.VI.1985)], Olympos Mt. [Prionia (21.V.1981), Stavros (20.V.1981)]. Total: 5 specimens.

Genus *Tritomegas* Amyot et Serville 1843 (Figure 2)

6.*Tritomegas bicolor* (Linnaeus 1758)

Epirus [Vryssochori (28.V.1981)], Voeotia [Arachova (17.III.1981)]. Total: 2 specimens.

7.*Tritomegas sexmaculatus* (Rambur 1839)

Olympos Mt. [Stavros (20.V.1981)]. Total: 1 specimen.


**Family PENTATOMIDAE**



**Subfamily Asopinae**


Genus *Arma* Hahn 1832 (Figure 4)

8.*Arma insperata* (Horváth 1899)

Florina [Vernon Mt-Kalo Nero (21.VIII.1986)]. Total: 1 specimen.

Genus *Jalla* Hahn 1832 (Figure 4)

9.*Jalla dumosa* (Linnaeus 1758)

Doris [Skaloula (1.VI.1985)], Fokis [Elaion (22.VII.1983)], Kozani [Vourinos Mt. 18.VIII.1983)]. Total: 3 specimens.

Genus *Picromerus* Amyot et Serville 1843 (Figure 4)

10.***Picromerus bidens**** (Linnaeus 1758) (Figure 3)

Florina [Vernon Mt-Kalo Nero (30.VII.1982)]. Total: 3 specimens.

**Figure 3 insects-13-00749-f003:**
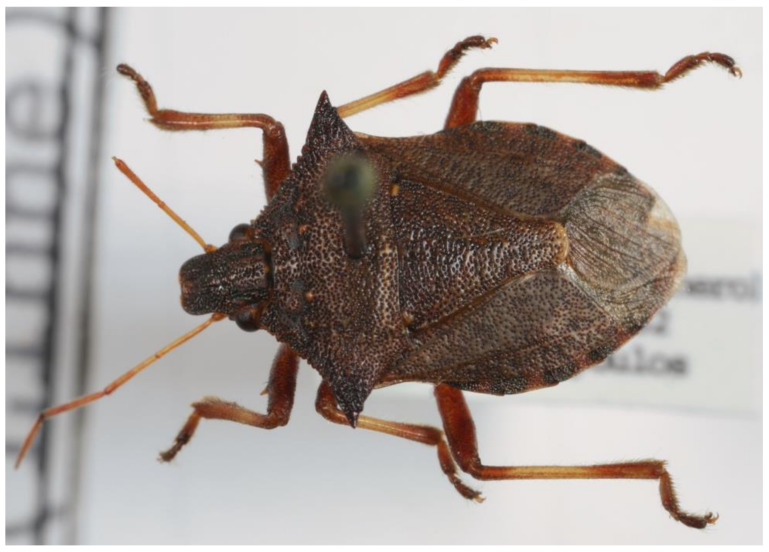
*Picromerus bidens* (Collector S. Drosopoulos, Vernon Mt.-Kalo Nero 30.VII.1982, posited in S. Drosopoulos historical collection. Photo A. Tsagkarakis 12.VIII.2022).

11.*Picromerus conformis* (Herrich-Schaeffer 1841)

Doris [Skaloula (27.IX.1983)], Florina [Psarades (18.VIII.1986)]. Total: 2 specimens.

Genus *Troilus* Stål 1867 (Figure 4)

12.*Troilus luridus* (Fabricius 1775)

Olympos Mt. [Stavros (20.V.1981)]. Total: 1 specimen.

Genus *Zicrona* Amyot et Serville 1843 (Figure 4)

13.*Zicrona caerulea* (Linnaeus 1758)

Attiki [Avlon (11.V.1980)], Doris [Monastirakion (18.VIII.1981)]. Total: 2 specimens.

**Figure 4 insects-13-00749-f004:**
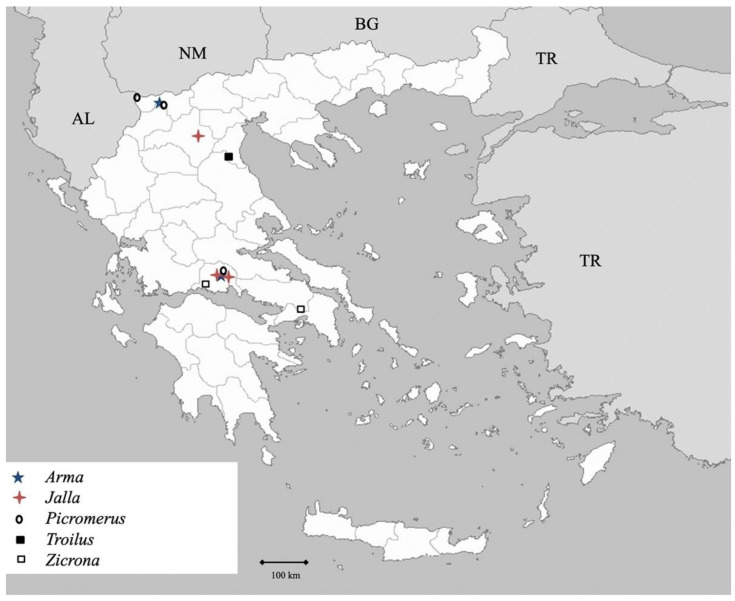
Map showing collecting sites of species of genera *Arma, Jalla, Picromerus, Troilus* and *Zicrona* during the present study in Greece.


**Subfamily Pentatominae**


Genus *Acrosternum* Fieber 1860 (Figure 6)

14.*Acrosternum heegeri* (Fieber 1861)

Doris [Eratini (14.VII.1978)], Zakynthos [Volimes (28.VI.1988)], Athos [Karyes (5.VIII.1986)]. Total: 7 specimens.

15.*Acrosternum millierei* (Mulsant et Rey 1866)

Attiki [Marathon (11.VII.1984)], Chios [Chora (5.IV.1979)], Doris [Agios Nikolaos (20.VII.1986)], Kynouria [Astros (31.VIII.1984)]. Total: 9 specimens.

Genus *Aelia* Fabricius 1803 (Figure 6)

16.*Aelia acuminata* (Linnaeus 1758)

Aetoloakarnania [Messolonghion (17.V.1990)], Amphilochia [Anixiatiko (20.VIII.1985)], Arkadia [Menalon Mt. (20.VI.1986)], Attiki [Amaroussion (24.V.1978, 27.IX.1978), Avlon (4.VII.1978, 16.VII.1979), Kifissia (10.I.1978, 8.II.1978), Parnis Mt(1.V.1983)], Chalkidiki [Valti (8.VIII.1985)], Chios [Chora (5.IV.1979, 20.IX.1984)], Doris [Kallion (12.V.1980), Skaloula (24.VII.1977, 28.IV.1978, 20.VI.1978, 12.VII.1978, 17.IX.1978, 25.X.1978, 27.III.1979, 29.IV.1979, 5.X.1979, 2.XII.1979, 16.VII.1987)], Epirus [Vryssochori (26.V.1981, 27.V.1981)], Evia [Oreoi (20.IV.1980)], Evrytania [Megalo Chorio (10.VIII.1986)], Drama [Falakron Mt. (13.VI.1982)], Florina [Kotas (15.VIII.1979, Vernon Mt. (30.VII.1982), Vigla (30.VIII.1983), Florina (15.VIII.1979)], Fokis [Galaxidi (15.IV.1980)], Fthiotis [Anthili (12.VIII.1979), Malessina (3.IV.1979), Oiti Mt. (9.VI.1984, 30.XI.1984, 15.VI.1985)], Grevena [Anoixis (17.VIII.1983, 30.VI.1984)], Ikaria [Gyaliskari (15.VII.1981), Raches (14.VII.1981)], Ioannina [Aristi (13.VIII.1986, 23.VIII.1986), Ioannina (29.V.1981)], Karditsa [Artessiano (1.X.1981)], Konitsa [Aoos Riv. (29.V.1981)], Lakonia [Mystras (30.IV.1985)], Lesvos [Sykamnia (16.VI.1987)], Messinia [Artemissia (29.IV.1985), Dorion (1.V.1985)], Naxos [Apeiranthos (16.VI.1981)], Olympos Mt. (13.VIII.1979), Parnassos Mt. [National Park (13.VII.1985)], Paros [Paroikia (18.VI.1981)], Pella [Vryta (14.VIII.1979)], Pieria [Poroi (12.VIII.1979, 14.VIII.1980)], Pindos Mt. [Miliotades (5.IX.1980, 29.IX.1980, 25.V.1981), Katara (21.VIII.1983)], Rodopi Mt. [Elatia (11.VI.1982, 26.VII.1982, 13.VIII.1985)], Rhodos [Dimilia (31.V.1990)], Strymonas Riv. (1.VI.1982), Thessaloniki [Cedron Hills (26.XII.1979, 31.V.1982, 14.X.1982), Plagiarion (23.V.1981)], Trikala [Orthovounion (30.IX.1981)], Vegoritis Lake (14.VIII.1979), Xanthi [Porto Lagos (9.VI.1982, 23.VIII.1983)], Zakynthos [Keri (29.VI.1988)]. Total: 119 specimens.

17.*Aelia albovittata* (Fieber 1868)

Chios [Armolia (20.VI.1987)], Samos [Pythagorion (23.VI.1987)]. Total: 3 specimens.

18.***Aelia germari**** (Küster 1852) (Figure 5)

Evros [Metaxades (2.VI.1982)]. Total: 1 specimen.

**Figure 5 insects-13-00749-f005:**
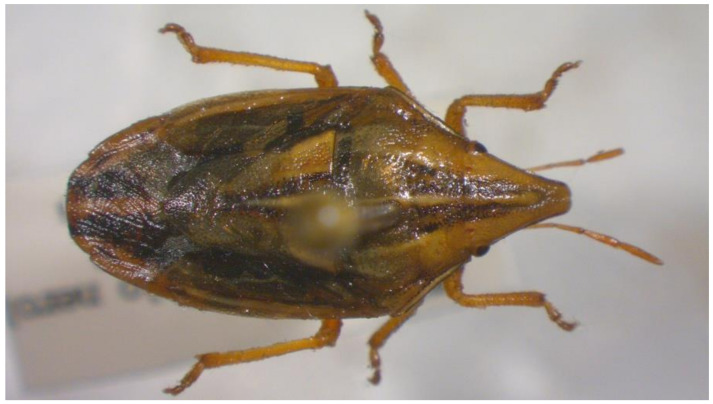
*Aelia germari* (Collector S. Drosopoulos, Metaxades, 2.VI.1982, posited in S. Drosopoulos historical collection. Photo A. Tsagkarakis, 12.III.2021).

19.*Aelia klugii* (Hahn 1831)

Florina [Vernon Mt. Kalo Nero (30.VII.1982, 12.VI.1982)], Rodopi [Vathyrhemma (12.VI.1982)]. Total: 5 specimens.

20.*Aelia rostrata* (Boheman 1852)

Attiki [Avlon (26.V.1978, 11.V.1979), Parnis Mt. (29.V.1985)], Doris [Ghiona (23.VII.1977, 16.VII.1878)], Drama [Potami (12.VI.1982)], Epirus [Vryssochori (28.V.1981)], Evros [Metaxades (2.VI.1982), Evros Riv. (Delta 6.VI.1982)], Florina (15.VIII.1979, Vernon Mt.-Kalo Nero (21.VII.1983)), Fthiotis [Anthili (12.VIII.1980)], Konitsa [Aoos Riv. (30.V.1981)], Kozani [Vourinos Mt. (18.VIII.1985)], Messinia [Artemissia (29.IV.1985), Dorion (1.V.1985)], Oiti Mt. (1.VII.1984, 15.VI.1985)], Olympos Mt. [Stavros (20.V.1981)], Parnon Mt. (18.VI.1986), Prionia (30.V.1982), Rhodos [Plymiri (29.VI.1987)], Rodopi Mt. [Elatia (19.VII.1983)]. Total: 38 specimens.

21.*Aelia virgata* (Klug 1841)

Evros [Metaxades (2.VI.1982)], Trikala [Aghiofilo (22.VII.1989)]. Total: 2 specimens.

Genus *Antheminia* Mulsant et Rey 1866 (Figure 6)

**Figure 6 insects-13-00749-f006:**
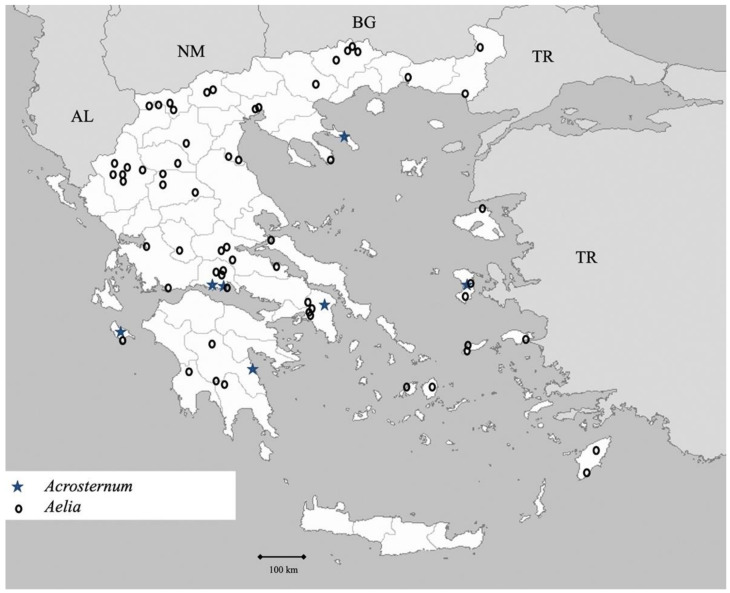
Map showing collecting sites of species of genera *Acrosternum* and *Aelia* during the present study in Greece.

22.*Antheminia lunulata* (Goeze 1778)

Fthiotis [Kalamakion (12.VII.1979)], Pella [Vryta (14.VIII.1979)], Vegoritis Lake (14.VIII.1979). Total: 3 specimens.

Genus *Apodiphus* Saunders 1877 (Figure 7)

23.*Apodiphus amygdali* (Germar 1817)

Attiki [Mati (20.VI.1977), Amaroussion (30.VIII.1985)], Doris [Skaloula (5.X.1980)]. Total: 7 specimens.

Genus *Brachynema* Mulsant et Rey 1852 (Figure 7)

24.*Brachynema cinctum* (Fabricius 1775)

Attiki [Schinias (1.VII.1982)]. Total: 1 specimen.

Genus *Carpocoris* Kolenati 1846 (Figure 7)

25.*Carpocoris (Carpocoris) fuscispinus* (Boheman 1851)

Attiki [Kifissia (30.III.1978), Marathon (14.V.1985), Parnis Mt. (13.VII.1984, 12.VII.1985)], Doris [Skaloula (24.VII.1977, 15.X.1979)], Epirus [Vryssochori (27.V.1981)], Evia [Agios Georgios (21.VI.1980), Aedipsos (23.VI.1980)], Kozani [Vourinos Mt. (17.VI.1985, 22.VII.1989)], Parnon Mt. [Kastanitsa (18.VI.1986), Kosmas (14.VII.1982, 11.VI.1985)], Messinia [Kazarma (4.VII.1984)], Olympos Mt. [Prionia (13.VIII.1979)], Paros [Paroikia (18.VI.1981)], Rhodos [Emponas (30.V.1990), Petaloudes (29.V.1990)], Samos [Neochorion (24.VI.1987), Pyrgos (24.VI.1987)], Oiti Mt. (9.VIII.1986), Thessaloniki [Cedron Hills (14.XI.1982)], Vegoritis Lake (14.VIII.1979). Total: 38 specimens.

26.*Carpocoris (Carpocoris) mediterraneus mediterraneus* (Tamanini 1958)

Evros Riv. [Delta (7.VI.1982)], Naxos [Moutsouna (15.III.1982)]. Total: 4 specimens.

27.*Carpocoris (Carpocoris) pudicus* (Poda 1761)

Attiki [Kifissia (30.III.1978), Parnis Mt. (29.V.1985)], Doris [Skaloula (20.VII.1978, 5.X.1980)], Florina [Vevi (24.VIII.1983)], Rodopi Mt. [Betoula Forest (22.V.1983)], Konitsa [Aoos Riv. (30.V.1981)], Kozani [Vourinos Mt. (17.VI.1985)], Vegoritis Lake (14.VIII.1979). Total: 10 specimens.

Genus *Chlorochroa* Stål 1872 (Figure 7)

28.*Chlorochroa* sp. Stål 1872

Olympos Mt. [Stavros (20.V.1981)]. Total: 1 specimen.

Genus *Codophila* Mulsant et Rey 1866 (Figure 8)

29.*Codophila varia* (Fabricius 1787)

Aetoloakarnania [Babini (23.VI.1986), Evinos Riv. (22.VI.1986)], Arkadia [Platanos (18.VI.1985), Xeropigado (9.V.1985)], Attiki [Parnis Mt. (13.VII.1984), Schinias (1.VII.1982)], Avlis [Vahty (16.VI.1979)], Doris [Skaloula (18.VII.1978, 20.VII.1978, 17.IX.1978, 15.VIII.1983, 1.VI.1985)], Evia [Agios Georgios (21.VI.1980)], Fokis [Eleon (17.VIII.1979)], Ikaria [Gyaliskari (15.VII.1981)], Ilia [Krestena (16.V.1985)], Kephallinia [Aenos Mt. (23.VI.1988)], Kynouria [Astros (5.VII.1984)], Messinia [Kazarma (4.VII.1984)], Parnassos Mt. (13.VII.1985), Rhodos [Emponas (30.V.1990), Malonas (30.V.1990)], Samos [Psili Ammos (22.VI.1987), Pythagorion (23.VI.1987)], Aoos Riv. (30.V.1981), Giona (23.VII.1982), Oiti Mt. (30.IX.1984), Parnon Mt. (18.VI.1986), Vegoritis Lake (14.VII.1979). Total: 43 specimens.

Genus *Dolycoris* Mulsant et Rey 1866 (Figure 8)

30.*Dolycoris baccarum* (Linnaeus 1758)

Achaia [Rion (9.VI.1980)], Aetoloakarnania [Babini 23.VI.1986)], Attiki [Amaroussion (2.III.1978), Kifissia (30.III.1978), Parnis Mt. (13.VII.1984, 12.VII.1985, 7.VII.1987)], Doris [Skaloula (15.VII.1978, 9.VI.1979, 29.VI.1979, 5.X.1980)], Epirus [Vryssochori (26.V.1981)], Evia [Edipsos (23.VI.1980)], Evros [Metaxades (2.VI.1982)], Florina (15.VIII.1979)], Fokis [Amfissa (19.IX.1978), Galaxidi (15.IV.1980)], Grevena [Anoixis (31.VII.1984)], Ilia [Kyllini (21.VII.1982)], Kephallinia [Aenos Mt. (23.VI.1988)], Konitsa [Aoos Riv. (29.V.1981)], Lakonia [Taygetos Mt. (30.IV.1985)], Messinia [Kazarma (4.VII.1984)], Kozani [Vourinos Mt. (28.V.1982)], Naxos [Moutsouna (16.VI.1981)], Oiti Mt. (30.IX.1984), Olympos Mt. [Stavros (13.VIII.1989, 20.V.1981), Sykaminos (30.V.1982)], Paros [Monastirion (17.VI.1981)], Pindos [Katara (4.IX.1980), Milies 20.VIII.1985)], Rhodos [Nea Afantou (28.VI.1987), Salakos (30.V.1990)], Rodopi Mt. [Elatia (25.VII.1982), Rodopi Mt. (27.VII.1982, 10.VIII.1985)], Samos [Pyrgos (14.VI.1987)], Parnon Mt. (18.VI.1986), Vourinos (28.V.1982)]. Total: 57 specimens.

Genus *Dyroderes* Spinola 1837 (Figure 8)

31.*Dyroderes umbraculatus* (Fabricius 1775)

Doris [Skaloula (28.IV.1981)], Epirus [Vryssochori (26.V.1981)], Messinia [Dorion (1.V.1985)], Pindos [Greveniti (6.IX.1980)]. Total: 9 specimens.

Genus *Eurydema* Laporte 1833 (Figure 8)

32.*Eurydema (Eurydema) eckerleini* (Josifov, 1961)

Syros [near Poseidonia (10.VI.2005)], Attiki [Ymitos Mt. (2.VIII.2005), Alimos (15.VII.2012), Votanikos (20.VII.2012)], Tinos [near Triantaros (29.VII.2016)]. Total: 100 specimens.

33.*Eurydema (Horvatheurydema) fieberi*, (Schummel 1837)

Attiki [Kifissia (15.IV.1974)], Doris [Giona (16.VII.1978), Skaloula (3.V.1989)], Parnon Mt. (30.VI.1989), Kozani [Vourinos Mt. (22.VII.1989)]. Total: 5 specimens.

34.*Eurydema (Eurydema) oleracea* (Linnaeus 1758)

Creta [Rethymnon-Myloi (8.VIII.1985)], Epirus [Vryssohori (26.V.1981)], Florina [Vernon Mt. (30.VII.1982), Vevi (30.VIII.1983)], Kozani [Vourinos Mt. (27.V.1982, 18.VIII.1983, 28.VI.1984, 18.VIII.1985)], Messinia [Artemissia (29.IV.1985)], Olympos Mt. [Prionia (13.VIII.1979, 20.V.1981, 21.V.1981), Stavros (12.V.1990)], Pindos [Miliotades (29.IX.1980), Panagia-Katara (25.V.1981)], Rodopi Mt. [Krya Vryssi (25.VII.1982), Rodopi Mt. (23.V.1983, 12.VIII.1985)], Oiti Mt. (30.IX.1984). Total: 40 specimens.

35.*Eurydema (Eurydema) ornata* (Linnaeus 1758)

Achaia [Bouboukas (21.VI.1986)], Aetoloakarnania [Mytikas-Astakos (23.VI.1986)], Arkadia [Tripolis (19.VI.1986)], Attiki [Amaroussion (27.IX.1978), Avlon (16.X.1979, 7.VIII.1987), Kifissia (9.VIII.1977, 30.III.1978, 11.VII.1980, 25.VIII.1980), Parnis Mt. (13.VII.1984)], Avlis [Vathy (3.VII.1978)], Doris [Agioi Pantes (26.V.1980), Skaloula (13.VII.1977, 13.VIII.1977, 12.VI.1978, 17.IX.1978, 19.VIII.1979, 30.V.1990)], Epirus [Vryssochori (27.V.1981)], Evia [Agios Georgios (21.VI.1980)], Florina [Kotas (15.VIII.1979)], Fthiotis [Malessina (21.III.1979)], Grevena [Anoixis (18.VIII.1983)], Kozani [Vourinos Mt. (22.VII.1979, 18.VIII.1983, 22.VII.1989)], Lesvos [Sykaminea (16.VI.1987)], Parnassos Mt. [National Park (13.VII.1985)], Pieria [Poroi (12.VIII.1979), Varikon (23.VII.1982)], Rhodos [Apolakkia (30.VI.1987), Dimilia (31.V.1990), Petaloudes (29.V.1990)], Samos [Agios Konstantinos (30.VI.1987)], Trikala [Mourgani (22.VII.1989)], Voeotia [Agia Paraskevi (8.VI.1980)], Ioannina (29.V.1981), Oiti Mt. (1.VII.1984, 15.VI.1985), Parnon Mt. (18.VI.1986, 30.VI.1989). Total: 75 specimens.

36.*Eurydema (Horvatheurydema) rugulosa* (Dohrn 1861)

Lesvos [Sykaminea (16.VI.1987)], Samos [Chora (25.VI.1987), Pyrgos (24.VI.1987)]. Total: 6 specimens.

37.*Eurydema (Rubrodorsalium) spectabilis* (Stichel 1960)

Crete [Agios Vassilios-Heracleon (10.VII.1985)]. Total: 12 specimens.

38.*Eurydema (Rubrodorsalium) ventralis* (Kolenati 1846)

Doris [Marathias 1.V.1979)]. Total: 24 specimens.

Genus *Eysarcoris* Hahn 1834 (Figure 10)

39.***Eysarcoris aeneus**** (Scopoli 1763) (Figure 9)

Rodopi Mt. [Sidironero (23.V.1983)]. Total: 1 specimen.

**Figure 9 insects-13-00749-f009:**
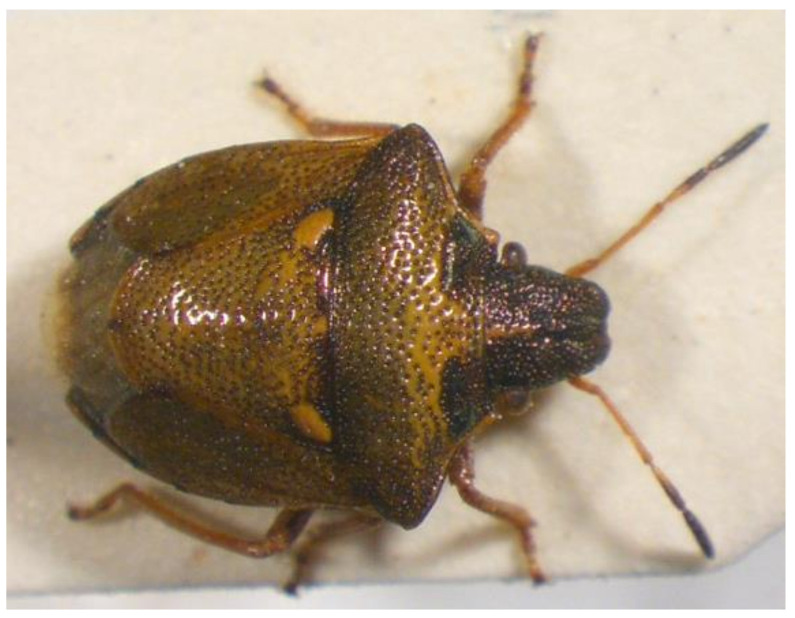
*Eysarcoris aeneus* (Collector S. Drosopoulos, Sidironero, 23.V.1983, posited in S. Drosopoulos historical collection. Photo A. Tsagkarakis, 12.III.2021).

40.*Eysarcoris ventralis* (Westwood, 1837)

Aetoloakarnania [Loutrakion-Vonitsa (22.VIII.1985)], Attiki [Amarousion (12.VII.1977), Avlon 4.VIII.1978), Kifissia (22.VI.1977), Schinias (1.VII.1982)], Crete [Rethymnon-Myloi (8.VII.1985)], Delta Acheloou (21.VII.1980), Delta Aliakmonos (19.VIII.1983), Doris [Kallion (18.VII.1978, 7.X.1979), Monastirakion (29.VII.1982), Skaloula (12.VII.1978)], Fokis [Amfissa (19.IX.1978)], Fthiotis [Anthili (26.VII.1977, 12.VIII.1980), Arkitsa (11.VIII.1980)], Ikaria [Gyaliskari (15.VII.1981)], Ioannina [Aristi (13.VIII.1985), Ioannina (20.VIII.1985)], Karpathos [Arkassa (3.VI.1990)], Korinthia [Kokoni (3.VIII.1977)], Messinia [Artemissia (11.VI.1985)], Olympos Mt. [Stavros (20.V.1981)], Paros [Paroikia (18.VI.1981)], Pindos [Katara (4.IX.1980), Miliotades (29.IX.1980)], Rhodos [Salakos (30.V.1990)], Samos [Agios Konstantinos (27.VI.1987), Psili Ammos (22.VI.1987)], Xanthi [Porto Lagos (9.VI.1982)], Loutra Kaiafa (3.VII.1984). Total: 49 specimens.

Genus *Halyomorpha* Mayr 1864 (Figure 10)

41.*Halyomorpha halys* (Stål 1855)

Attiki [Votanikos (2.II.2015, 10.X.2016, 6.IX.2017), Alimos (20.V.2015), Kifissia (25.V.2015)], Crete [Chania (18.X.2015)], Kastellorizo [around village (15.VIII.2015)]. Total: 25 specimens.

Genus *Holcogaster* Fieber 1860 (Figure 10)

42.*Holcogaster fibulata* (Germar 1831)

Attiki [Marathon (11.VII.1984)], Doris [Agios Nikolaos (20.VII.1986)]. Total: 2 specimens.

Genus *Holcostethus* Fieber 1860 (Figure 10)

43.*Holcostethus albipes* (Fabricius 1781)

Kynouria [Astros (31.VII.1984)]. Total: 1 specimen.

44.*Holcostethus sphacelatus* (Fabricius 1794)

Epirus [Vryssochori (26.V.1981)], Konitsa [Aoos Riv. 29.V.1981)], Olympos Mt. [Prionia (21.V.1981)], Rodopi Mt. [Virgin Wood 23.V.1983)]. Total: 7 specimens.

Genus *Mustha* Amyot et Serville 1843 (Figure 10)

45.*Mustha spinulosa* (Lefebvre 1831)

Attiki [Avlon (26.VII.1979), Kifissia (16.VI.1978, 25.VII.1980)], Doris [Skaloula (18.V.1989)], Trikala [Mourgani (19.VI.1985)]. Total: 6 specimens.

**Figure 10 insects-13-00749-f010:**
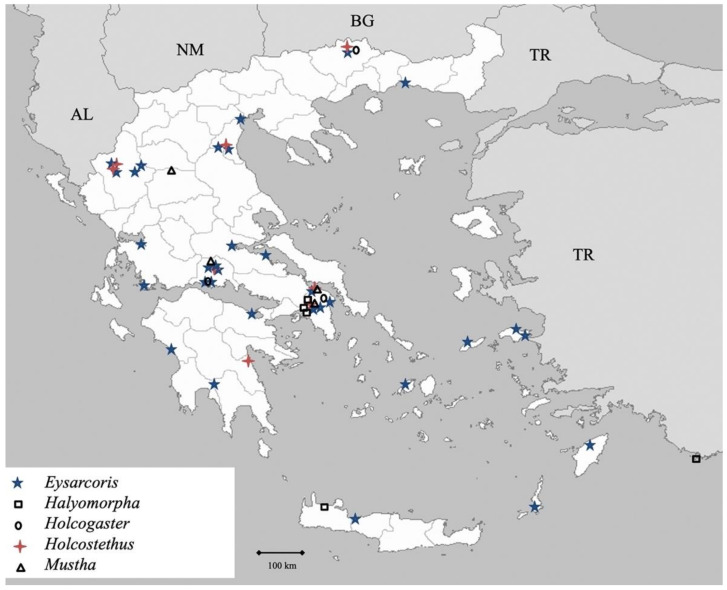
Map showing collecting sites of species of genera *Eysarcoris, Halyomorpha, Holcogaster, Holcostethus* and *Mustha* during the present study in Greece.

Genus *Neostrachia* Dallas 1851 (Figure 13)

46.*Neostrachia bisignata* (Walker 1867)

Doris [Monastirakion (29.VII.1982)], Kynouria [Astros (24.VIII.1988)], Volos [Amaliapolis (1.IX.1990)]. Total: 3 specimens.

Genus *Neottiglossa* Kirby 1837 (Figure 13)

47.*Neottiglossa bifida* (Costa 1847)

Achaia [Bouboukas (21.VI.1986)], Aetoloakarnania [Messolonghion-Tourlida (20.VII.1990), Mytikas-Astakos (23.VI.1986)], Doris [Kallion (14.VIII.1977), Monastirakion (18.VIII.1981), Skaloula (24.VII.1977, 13.VII.1978, 9.VI.1979, 1.VI.1985)], Messinia [Menina (30.IV.1985), Artemissia 29.VI.1985), Kazarma (4.VII.1984)], Konitsa [Aoos-Monastiri (30.V.1981)], Lakonia [Mystras (30.IV.1985)], Naxos [Apeiranthos (16.VI.1981)], Pieria [Poroi (14.VIII.1980)], Thessaloniki [Plagiarion (23.V.1981)]. Total: 22 specimens.

48.*Neottiglossa flavomarginata* (Lucas 1849)

Kozani [Vourinos Mt. (29.VI.1984)]. Total: 6 specimens.

49.*Neottiglossa leporina* (Herrich-Schaeffer 1830)

Doris [Krokylion (19.VII.1978), Mavrolithari (8.VIII.1986)], Drama [Potami (12.VI.1982)], Epirus [Vryssochori (26.V.1981)], Florina [Vernon Mt. Kalo Nero (30.VII.1982), Florina (1.X.1983)], Konitsa [Aoos-Monastiri (10.V.1981)], Kozani [Vourinos Mt. (8.VI.1984)], Pieria [Poroi (14.VIII.1980)], Rodopi Mt. [Sidironero (23.V.1983), Vathyrhemma (26.VII.1982)], Oiti Mt. (30.IX.1984, 15.VI.1985)]. Total: 22 specimens.

50.***Neottiglossa lineolata**** (Mulsant et Rey 1852) (Figure 11)

Doris [Skaloula (18.VII.1978)], Evia [Agios Georgios (21.VI.1980), Gialtra (23.VI.1980)], Konitsa [Aoos-Monastiri (30.V.1981)], Lesvos [Antissa (17.VI.1987)], Xanthi [Porto Lagos (9.VI.1982)]. Total: 8 specimens.

**Figure 11 insects-13-00749-f011:**
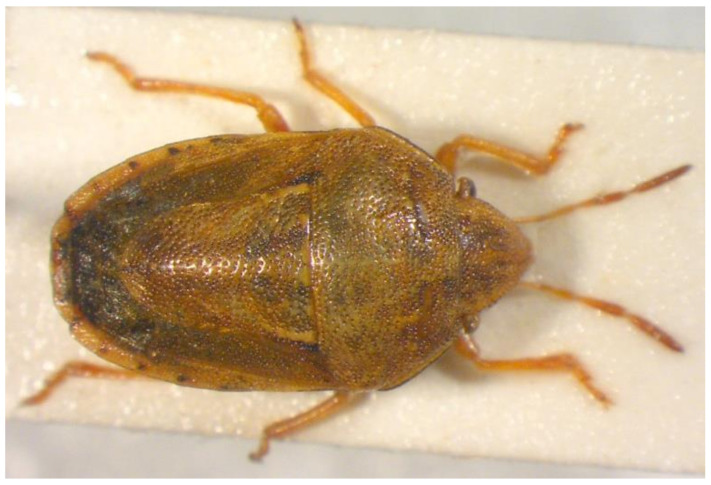
*Neottiglossa lineolata* (Collector S. Drosopoulos, Porto Lagos, 9.VI.1982, posited in S. Drosopoulos historical collection. Photo A. Tsagkarakis, 12.III.2021).

51.***Neottiglossa pusilla**** (Gallen 1789) (Figure 12)

Rodopi Mt. [Vathyrhemma (26.VII.1982)]. Total: 2 specimens.

**Figure 12 insects-13-00749-f012:**
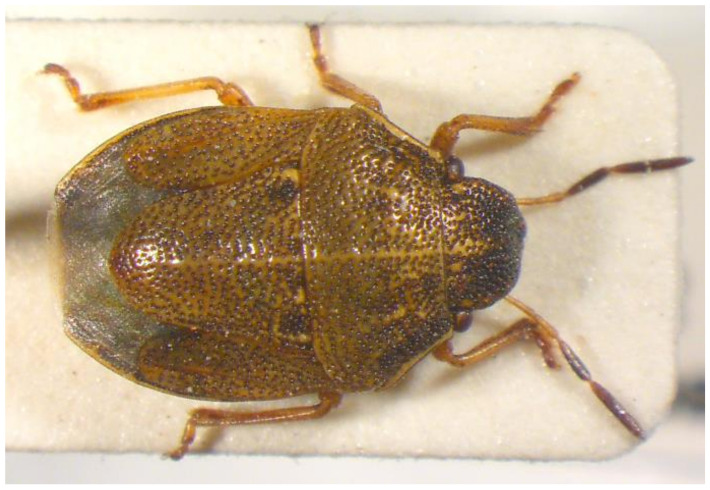
*Neottiglossa pusilla* (Collector S. Drosopoulos, Vathyrhemma, 26.VII.1982, posited in S. Drosopoulos historical collection. Photo A. Tsagkarakis, 12.III.2021).

Genus *Nezara* Amyot et Serville 1843 (Figure 13)

52.*Nezara viridula* (Linnaeus 1758)

Attiki [Chalandri (3.VII.1987), Kifissia (9.VIII.1977, 18.VIII.1978, 2.VII.1979, 29.X.1984, 6.II.1986), Mati (20.VI.1977)]. Crete [Heracleon-Agios Vassilios (10.VII.1985)], Paros [Paroikia (18.IX.1981)]. Total: 13 specimens.

Genus *Palomena* Mulsant et Rey 1866 (Figure 13)

**Figure 13 insects-13-00749-f013:**
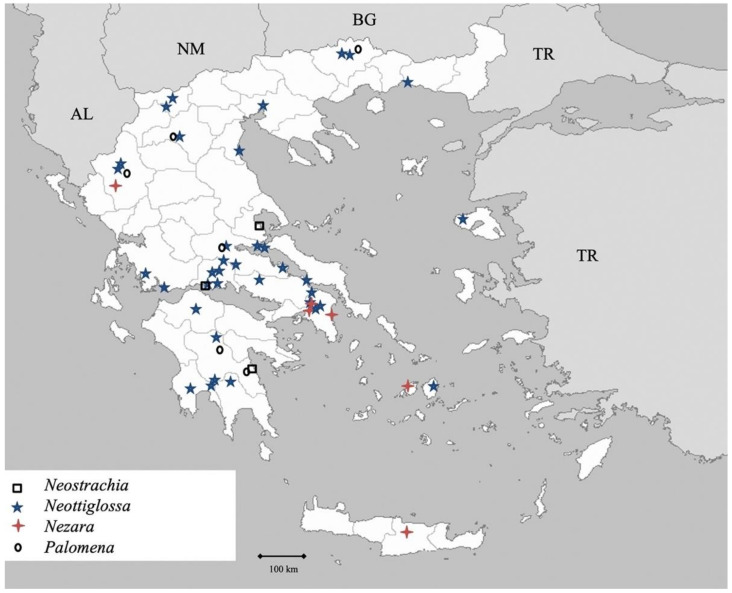
Map showing collecting sites of species of genera *Neostrachia, Neottiglossa, Nezara* and *Palomena* during the present study in Greece.

53.*Palomena prasina* (Linnaeus 1761)

Kozani [Vourinos Mt. (28.V.1982)], Oiti Mt. (30.IX.1984), Parnon Mt. [Kastanitsa (19.VII.1982, 15.IX.1983)], Pindos [Miliotades (29.IX.1980)], Rodopi Mt. [Betula Forest (22.V.1983)], Tripolis [Tripolis (19.VI.1986)]. Total: 10 specimens.

Genus *Pentatoma* Olivier 1789 (Figure 14)

54.*Pentatoma (Pentatoma) rufipes* (Linnaeus 1758)

Florina [Vernon Mt.-Kalo Nero (21.VII.1983, 21.VIII.1986), Vigla (30.VIII.1983)], Rodopi Mt. [Betula Forest (25.VII.1992), Sidironero (27.VII.1992)]. Total: 10 specimens.

Genus *Peribalus* Mulsant et Rey, 1866 (Figure 14)

55.*Peribalus strictus strictus* (Fabricius 1803)

Attiki [Avlon (4.VIII.1978), Kifissia (30.VII.1978, 8.X.1987)], Avlis [Vathy (3.VII.1978)], Crete [Agios Vassilios-Heracleon (25.IV.1975)], Doris [Skaloula (18.VII.1978, 17.IX.1978, 7.VIII.1986)], Florina [Kotas (15.VIII.1979)], Fthiotis [Malessina (21.III.1979)], Parnassos Mt. [National Park (13.VII.1985)], Rhodos [Salakos (30.V.1990)], Samos [Agios Konstantinos (27.VI.1987)]. Total: 16 specimens.

Genus *Piezodorus* Fieber 1861 (Figure 14)

56.*Piezodorus lituratus* (Fabricius 1794)

Attiki [Parnis-Mola (12.VII.1985)], Schinias [(16.V.1984)], Avlis [Vathy 13.IV.1978)], Doris [Eratini (4.III.1979, 8.III.1988), Skaloula (28.IV.1978, 12.VII.1978, 14.VII.1978, 30.X.1982)], Epirus [Vryssochori (26.V.1981)], Evia [Agios Georgios (21.VI.1980)], Evrytania [Megalo Chorio (10.VIII.1986)], Florina [Vernon Mt.-Kalo Nero (30.VII.1982)], Messinia [Menina (30.IV.1985)], Ilia [Killini Mt. (21.VII.1982)], Kerkyra [Tritsi (3.IV.1985)], Oaks [Rodopi (12.VIII.1985)], Parnon Mt., [Kastanitsa (14.VIII.1982, 15.IX.1983)], Ikaria [Raches (14.VII.1981)], Rhodos [Apolakkia (30.VI.1987), Emponas (30.VI.1990), Messanagros (29.VI.1987), Petaloudes (29.V.1990)]. Total: 31 specimens.

Genus *Rhaphigaster* Laporte 1833 (Figure 14)

57.*Rhaphigaster nebulosa* (Poda 1761)

Acheloos Riv. [Delta Acheloou (21.VII.1990)], Attiki [Kifissia (16.II.1978, 18.VIII.1978, 5.VI.1980, 12.IX.1983), Mati (28.VIII.1977)], Avlis [Vathy (5.V.1981)], Crete [Heracleon-Agios Vassilios (10.VII.1985)], Doris [Mavrolithari (8.VIII.1986), Skaloula (2.XI.1979, 27.IX.1983, 26.III.1984, 21.XII.1987)], Evros Riv. [Delta Evrou (7.VI.1982)], Aetoloakarnania [Nafpaktos (4.IX.1979)], Rhodos [Petaloudes (29.V.1990)]. Total: 17 specimens.

Genus *Sciocoris* Fallen 1829 (Figure 15)

58.*Sciocoris (Sciocoris) cursitans cursitans* (Fabricius 1794)

Doris [Skaloula (12.IV.1985)], Lesvos [Andissa (17.VI.1987)]. Total: 2 specimens.

59.*Sciocoris (Sciocoris) deltocephalus* (Fieber 1861)

Kephallinia [Aenos Mt. (23.VI.1988)], Paros [Paroikia (18.VI.1981)]. Total: 2 specimens.

60.*Sciocoris (Sciocoris) helferii* (Fieber 1851)

Aetoloakarnania [Mytikas-Astakos (23.VI.1986)], Arkadia [Xeropigado (9.V.1985)], Attiki [Parnis Mt.-Mola (12.VII.1985)], Ikaria [Gialiskari (15.VII.1981)], Kephallinia [Aenos Mt. (23.VI.1988)], Konitsa [Aoos-Monastiri (30.V.1981)], Kozani [Vourinos Mt. (29.VI.1984)], Aoos Riv. (14.VIII.1986)], Lesvos [Kalloni (18.VI.1987)], Pieria [Varikon (23.VII.1982)]. Total: 11 specimens.

61.*Sciocoris (Aposciocoris) macrocephalus* (Fieber 1851)

Arkadia [Xeropigado (9.V.1985)], Attiki [Avlon (26.V.1978, 4.VIII.1978, 4.VII.1979) Marathon (11.VII.1984, 14.V.1985)], Doris [Skaloula (1.VI.1985)], Fokis [Amfissa (2.V.1979)], Kephallinia [K. Katelios (22.VI.1988)], Kerkyra [Liapades (2.IV.1985)]. Total: 16 specimens.

62.*Sciocoris (Neosciocoris) maculatus* (Fieber 1851)

Kephallinia [Aenos Mt. (23.VI.1988)], Konitsa [Aoos-Monastiri (29.V.1981, 30.V.1981), Aoos Riv. (14.VIII.1986)], Olympos Mt. [Kryovryssi (30.V.1982)], Samothraki [Loutra (4.VI.1982)], Oiti Mt. (9.VIII.1986). Total: 8 specimens.

63.*Sciocoris (Sciocoris) sulcatus* (Fieber 1851)

Aetoloakarnania [Messolongion-Tourlida (20.VII.1990)], Attiki [Avlon (26.V.1978, 11.V.1979), Kifissia (25.VIII.1980), Parnis Mt. (29.V.1985)], Kephallinia [Aenos Mt. (23.VI.1986)], Korinthia [Derveni (15.V.1985)], Kozani [Vourinos Mt. (29.VI.1984)], Kynouria [Kastanitsa (20.V.1982)], Messinia [Dorion (1.V.1985)], Olympos Mt. [Prionia (21.V.1981)], Pindos [Milies (20.VIII.1985)], Rhodos [Kalathos (28.VI.1987), Nea Afantou (28.VI.1987), Petaloudes (29.V.1990)], Rodopi Mt. [Elatia (25.VII.1982)], Voeotia [Tsoukalades (12.V.1988)]. Total: 19 specimens.

Genus *Stagonomus* Gorski 1852 (Figure 15)

64.*Stagonomus (Stagonomus) amoenus* (Brulle 1832)

Crete [Heracleon-Agios Vassilios (25.IV.1985)], Konitsa [Aoos Riv. (29.V.1981), Aoos-Monastirion (30.V.1981)], Paros [Monastirion (17.VI.1981)], Rhodos [Petaloudes (29.V.1990)]. Total: 15 specimens.

65.*Stagonomus (Dalleria) bipunctatus* (Linnaeus 1758)

Evia [Agios Georgios (21.VI.1980)], Florina [Vermon Mt.-Kalo Nero (30.VII.1982)], Kozani [Vourinos Mt. (18.VIII.1985)]. Total: 7 specimens.

Genus *Staria* Dohrn 1860 (Figure 15)

66.*Staria lunata* (Hahn 1835)

Arkadia [Xeropigado (9.V.1985)], Attiki [Marathon (11.VII.1984)], Doris [Skaloula (15.VII.1978, 20.VII.1978, 17.IX.1978, 29.IV.1979)], Evros [Metaxades (2.VI.1982)], Fokida [Agia Euthymia (12.V.1988)], Konitsa [Aoos (27.V.1981)], Korinthia [Akrokorinthos (31.VII.1979)], Kozani [Vourinos Mt. (28.VI.1984)] Menalon Mt. (20.VI.1986), Parnassos Mt. [National Park (13.VII.1985), Parnassos Mt. (23.V.1980)], Rodopi Mt. [Vathyrhemma (26.VII.1982)]. Total: 36 specimens.


**Subfamily Podopinae**


Genus *Ancyrosoma* Amyot et Serville 1843 (Figure 16)

67.*Ancyrosoma leucogrammes* (Gmelin 1790)

Aetoloakarnania [Aetolikon (20.VII.1990), Messolonghion-Tourlida (16.V.1990), Kapsorachi-Trichonis (22.VI.1986)], Arkadia [Xeropigado (9.V.1985)], Arta [Louros Riv. (28.IX.1981)], Attiki [Amaroussion (27.IX.1978), Avlon (4.VII.1978, 12.VII.1979, 16.X.1979), Kifissia (14/31978, 11.VII.1980), Marathon (14.V.1985), Parnis Mt. (13.VII.1984), Schinias (1.VII.1982, 20.V.1983)], Avlis [Vathy (16.VI.1979)], Doris [Eratini (11.VI.1979), Ghiona (16.VII.1978), Kallion (18.VII.1978), Karoutes (18.IX.1978), Skaloula (13.VIII.1977, 18.VI.1978, 20.VII.1978, 17.IX.1978, 15.VIII.1983, 1.VI.1985)], Evia [Agios Georgios (21.VI.1980)], Fokis (Agia Euthymia (12.V.1988)], Ioannina (29.V.1981), Karditsa [Artessiano (1.X.1981)], Konitsa [Aoos Riv. (30.V.1981, 17.VI.1981)], Korinthia [Akrokorinthos (31.VII.1979), Derveni (15.V.1985),], Kynouria [Astros (13.VII.1982, 9.IX.1982)], Messinia [Kazarma (4.VII.1984)], Olympos Mt. [Prionia (13.VIII.1979)], Pieria [Poroi (14.VIII.1980)], Platanos [Parnon Mt. (1.VII.1985)], Thessaloniki [Cedron Hills (23.V.1981)]. Total: 57 specimens.

Genus *Derula* Mulsant et Rey 1856 (Figure 16)

68.*Derula flavoguttata* (Mulsant et Rey 1856)

Attiki [Schinias (23.VI.1986)], Doris [Giona (23.VII.1977, 16.VII.1978), Skaloula (18.VI.1978, 1.VI.1985)], Epirus [Vryssochori (26.V.1981)], Evia [Agios Georgios (21.VI.1980), Gialtra (23.VI.1980)], Florina [Vernon Mt.-Kalo Nero (30.VII.1982), Vevi (25.VIII.1983)], Fokis [Agia Euthymia (12.V.1988)], Korinthia [Derveni (15.V.1985)], Pindos [Vouchorina (13.V.1983)], Rhodos [Salakos (30.V.1990)], Samos [Pythagorion (23.VI.1987)], Trikala [Mourgani (18.VI.1985)]. Total: 23 specimens.

Genus *Graphosoma* Laporte 1833 (Figure 16)

69.*Graphosoma italicum italicum* (Müller 1766)

Aetoloakarnania [Aetolikon (20.VII.1990)], Attiki [Kifissia (9.VIII.1977, 10.I.1978), Schinias (23.IV.1986)], Crete [Rethymnon-Myloi (8.VII.1985)], Doris [Skaloula (14.VII.1978, 7.VIII.1986)], Evrytania [Megalo Chorio (10.VIII.1986)], Fokis [Amfissa (8.VI.1980)] Ioannina [Vikos Gorge (13.VIII.1986)], Konitsa [Aoos-Monastiri (3.V.1981)], Korinthia [Derveni (15.V.1985)], Kozani [Vourinos Mt. (28.VI.1984), (18.VIII.1985)], Lakonia [Mystras (30.IV.1986)], Lesvos [Sykaminea (16.VI.1987)], Olympos Mt. [Stavros (13.VIII.1980)], Pieria [Litochoron (13.VIII.1979)], Rhodos [Apolakkia (30.VI.1987)], Voeotia [Aliartos (12.V.1988)], Oiti Mt. (30.IX.1984, 15.VI.1985). Total: 28 specimens.

70.*Graphosoma semipunctatum* (Fabricius 1775)

Attiki [Amaroussion (27.IX.1978), Avlon (16.X.1979), Marathon (14.V.1985), Schinias (23.IV.1986)], Crete [Rethymnon-Agioi Apostoloi (30.VI.1993)], Doris [Skaloula (18.VI.1978)], Kephallinia [Myrtos (25.VI.1988)], Lakonia [Mystras (11.VI.1985)], Oidi Mt. (27.VI.1993), Samos [Neochorion (14.VI.1987)]. Total: 23 specimens.

Genus *Leprosoma* Baerensprung 1859 (Figure 16)

71.*Leprosoma inconspicuum* (Baerensprung 1859)

Doris [Kalion (5.V.1975)]. Total: 1 specimen.

Genus *Podops* Laporte 1833 (Figure 19)

72.*Podops (Opocrates) curvidens* (Costa 1843)

Aetoloakarnania [Loutrakion-Vonitsa (22.VIII.1985), Messolongion-Tourlida (20.VII.1990)], Doris [Eratini (11.VI.1979), Kalion (15.VII.1978, 5.III.1979, 7.X.1979), Monastirakion (29.IV.1981), Skaloula (2.XI.1979)], Pieria [Varikon (23.VII.1982)], Pindos [Korydallos (25.V.1981), Miliotates (5.IX.1980)]. Total: 14 specimens.

73.***Podops (Podops) inunctus**** (Fabricius 1775) (Figure 17)

Doris [Kalion (15.VII.1978)], Samothraki [Kamariotissa (5.VI.1982)]. Total: 4 specimens.

**Figure 17 insects-13-00749-f017:**
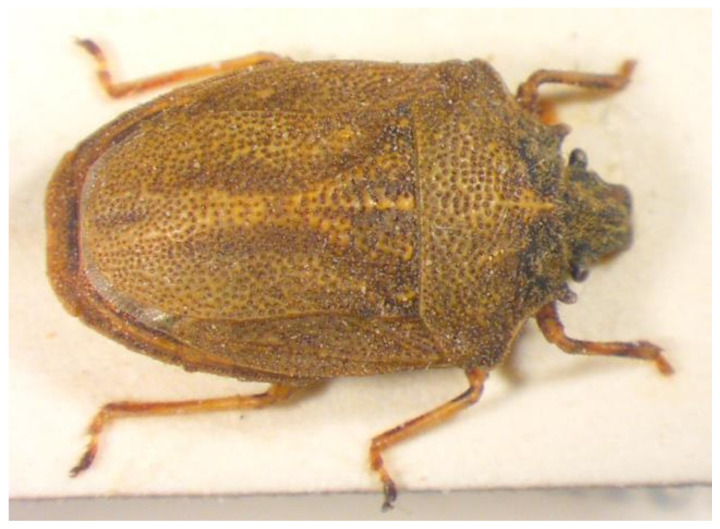
*Podops (Podops) inunctus* (Collector S. Drosopoulos, Kamariotissa, 5.VI.1982, posited in S. Drosopoulos historical collection. Photo A. Tsagkarakis, 12.III.2021).

74.*Podops (Opocrates) rectidens* Horvath 1883

Achaia [Bouboukas (9.VI.1980)], Aetoloakarnania [Mytikas-Astakos (23.VI.1986)], Arkadia [Kynouria-Astros (20.V.1982, 5.VII.1984)], Attiki [Schinias (1.VII.1982)], Doris [Eratini (11.VI.1979), Kalion (7.X.1979), Monastirakion (9.VI.1980, 25.X.1980)], Ikaria [Gialiskari (15.VII.1981)], Ilia [Pyrgos-Zacharo (3.VII.1984)], Samothraki [Kamariotissa (5.VI.1982)], Zakynthos [L. Keri (29.VI.1988)]. Total: 23 specimens.

Genus *Tarisa* Amyot et Serville 1843 (Figure 19)

75.***Tarisa pallescens**** (Jakovlev 1871) (Figure 18)

Olympos Mt. [Kryovryssi (30.V.1982)]. Total: 1 specimen.

**Figure 18 insects-13-00749-f018:**
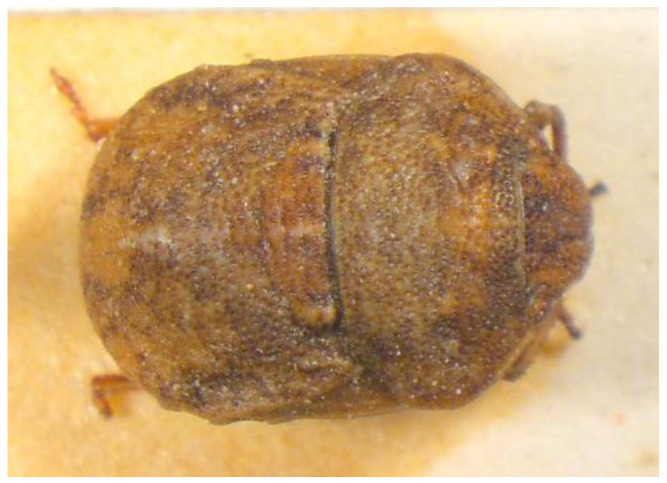
*Tarisa pallescens* (Collector S. Drosopoulos, Kryovryssi, 30.V.1982, posited in S. Drosopoulos historical collection. Photo A. Tsagkarakis, 12.III.2021).

Genus *Tholagmus* Stål 1860 (Figure 19)

**Figure 19 insects-13-00749-f019:**
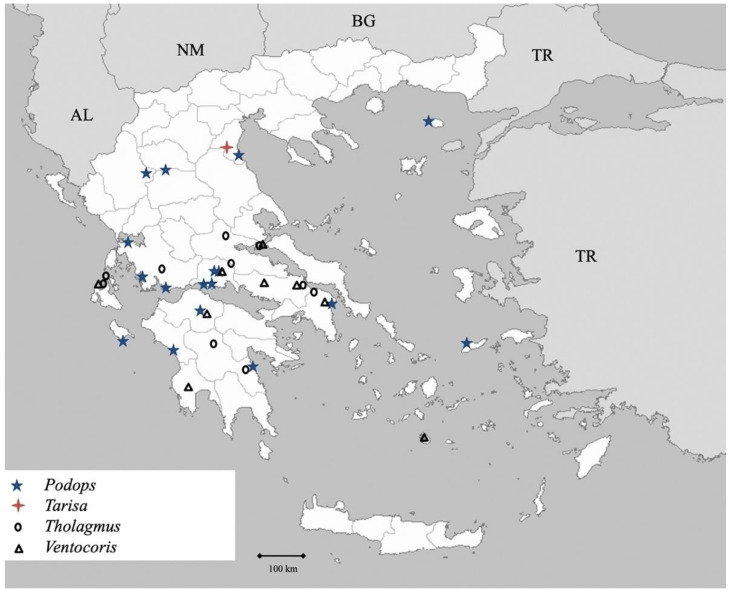
Map showing collecting sites of species of genera *Podops, Tarisa, Tholagmus* and *Ventocoris* during the present study in Greece.

76.*Tholagmus flavolineatus* (Fabricius 1798)

Arkadia [Charadros (17.VI.1986)], Attiki [Avlon (7.VI.1978, 4.VII.1979, 31.V.1989)], Avlis [Vathy (16.VI.1979)], Evia [Agios Georgios (21.VI.1980)], Fthiotis [Kalamakion (18.VIII.1979)], Kephallinia [Assos (25.VI.1988), Fiskardo(24.VI.1988)], Aetoloakarnania [Kapsorachi-Trichonis (22.VI.1986)], Menalon Mt. (20.VI.1986), Parnassos Mt. [National Park (13.VII.1985)]. Total: 18 specimens.

Genus *Ventocoris* Hahn 1843 (Figure 19)

77.*Ventocoris (Selenodera) achivus* (Horvath 1889)

Avlis [Vathy (16.VI.1979)]. Total: 2 specimens.

78.*Ventocoris (Ventocoris) rusticus* (Fabricius 1781)

Achaia [Kato Klitoria (20.VI.1986)], Attiki [Marathon (14.V.1985)], Avlis [Vathy (23.V.1979, 16.VI.1979)], Evia [Agios Georgios (21.VI.1980)], Fokis [Amphissa (8.VI.1980)], Kephallinia [Assos (25.VI.1988)], Messinia [Kazarma (4.VII.1984)], Santorini Island [Kamarion (5.V.1982)], Voeotia [Agia Paraskevi (23.VII.1980)]. Total: 23 specimens.


**Family PLATASPIDAE**



**Subfamily Plataspinae**


Genus *Coptosoma* Laporte 1833 (Figure 20)

79.*Coptosoma scutellatum* (Geoffroy 1785)

Achaia [Bouboukas (21.VI.1986)], Doris [Kallion (15.VII.1978), Krokylion (21.VI.1978, 19.VII.1978), Skaloula (18.VI.1978)], Florina [Vernon Mt.-Kalo Nero (21.VII.1983)], Ioannina [Voutsaras (24.VI.1984)], Kozani [Vourinos Mt. (18.VIII.1985)], Olympos Mt. [Prionia (13.VIII.1980)], Parnassos Mt. [National Park (13.VII.1985)], Pindos [Korydallos (25.V.1981)], Prespa [Bela Voda (13.X.1986)], Rodopi [Vathyrhemma (26.VII.1982)], Aoos Riv. (27.V.1981). Total: 32 specimens.


**Family SCUTELLERIDAE**



**Subfamily Eurygastrinae**


Genus *Eurygaster* Laporte 1833 (Figure 22)

80.*Eurygaster austriaca* (Schrank 1778)

Avlis [Vathy (7.VI.1978, 23.V.1979)], Fokis [Galaxidi (15.IV.1980)], Oiti Mt. (1.VII.1984). Total: 4 specimens.

81.*Eurygaster dilaticollis* (Dohrn 1860)

Rodopi Mt. (19.VII.1983), Oiti Mt. (1.VII.1984, 15.VI.1985), Vernon Mt. [Kalo Nero (30.VI.1982)]. Total: 6 specimens.

82.***Eurygaster hottentotta**** (Fabricius 1775) (Figure 21)

Kynouria [Kastanitsa (20.VII.1982)]. Total: 1 specimen.

**Figure 21 insects-13-00749-f021:**
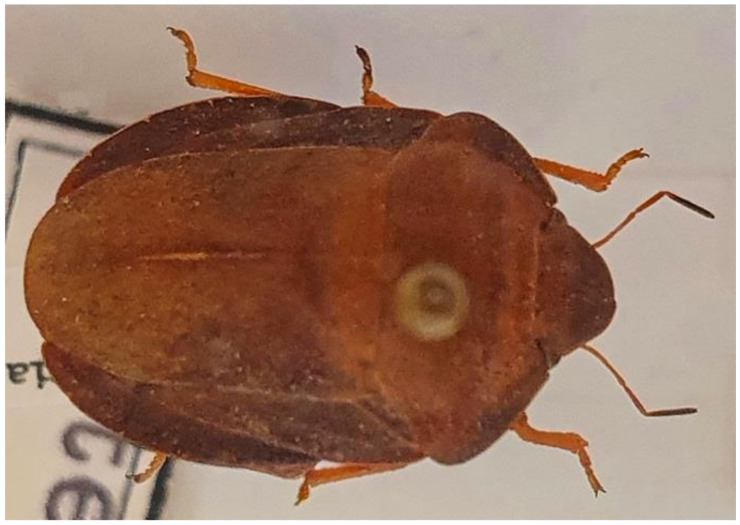
*Eurygaster hottentotta* (Collector S. Drosopoulos, Kastanitsa, 20.VII.1982, posited in S. Drosopoulos historical collection. Photo A. Tsagkarakis, 12.III.2021).

83.*Eurygaster integriceps* (Puton 1881)

Arkadia [Xeropigado (9.V.1985)], Attiki [Marathon (14.V.1978)], Avlis [Vathy (13.IV.1978)], Chios [Agios Georgios (6.IV.1979), Armolia (20.VI.1987)], Doris [Skaloula (29.IV.1979, 17.V.1981)], Epirus [Vryssochori (26.V.1981)], Grevena [Agioi Theodoroi (18.VI.1985), Anoixis (31.VII.1984)], Ikaria [Raches (14.VII.1981)], Ioannina (29.V.1981), Kozani [Vourinos Mt. (29.VI.1984)], Kynouria [Astros (20.V.1982)], Naxos [Apeiranthos (16.VI.1981)], Paros [Paroikia (18.VI.1981)], Samos [Agios Konstantinos (27.VI.1987), Pythagorion (23.VI.1987)]. Total: 31 specimens.

84.*Eurygaster maura* (Linnaeus 1758)

Doris [Monastirakion (29.VII.1982), Skaloula (15.VII.1978, 28.X.1981)], Epirus [Vryssochori (27.V.1981)], Evia [Agios Georgios (21.VI.1980)], Florina [Megali Prespa (30.VII.1982), Florina (15.VIII.1979)], Kastoria (13.V.1983), Kynouria [Astros (19.V.1982, 5.VII.1984)], Messinia [Kazarma (4.VII.1984)], Pindos [Korydallos (25.V.1981)], Rhodos [Archipolis (31.V.1990), Messanagros (29.IX.1987)], Rodopi Mt. (25.VII.1982, 10.VIII.1985), Xanthi [Porto Lagos (8.V.1982)], Zakynthos [Keri (29.VI.1988)]. Total: 21 specimens.

85.*Eurygaster testudinaria testudinaria* (Geoffroy 1785)

Kastoria (13.V.1983), Rodopi Mt. [Silli (23.VIII.1983)]. Total: 3 specimens.

Genus *Psacasta* Germar 1839 (Figure 22)

86.*Psacasta (Psacasta) exanthematica exanthematica* (Scopoli 1763)

Aetoloakarnania [Nafpaktos (22.VII.1978)]. Total: 1 specimen.


**Subfamily Odontoscelinae**


Genus *Irochrotus* Amyot et Serville 1843 (Figure 22)

87.*Irochrotus maculiventris* (Germar 1839)

Achaia [Bouboukas (21.VI.1986)], Arkadia [Tripolis (19.VI.1986)], Attiki [Avlon (26.V.1978)], Doris [Skaloula (1.VI.1985)], Evia [Agios Georgios (21.VI.1980), Mantoudi (28.X.2007)], Ioannina [Voutsaras (24.VI.1984)]. Total: 8 specimens.

Genus *Odontoscelis* Laporte 1833 (Figure 22)

88.*Odontoscelis (Odontoscelis) fuliginosa* (Linnaeus 1761)

Attiki [Avlon (11.V.1980), Parnis Mt. (29.V.1985)], Karpathos [Messochori (4.VI.1990)], Kephallinia [Aenos Mt. (23.VI.1988)]. Total: 7 specimens.

89.*Odontoscelis (Odontoscelis) lineola* (Rambur 1839)

Doris [Skaloula (17.IX.1978)] Arkadia [Charadros (17.VI.1989), Karoutes (18.IX.1978)], Vegoritis Lake (14.VIII.1979). Total: 4 specimens.


**Subfamily Odontotarsinae**


Genus *Odontotarsus* Laporte 1833 (Figure 22)

**Figure 22 insects-13-00749-f022:**
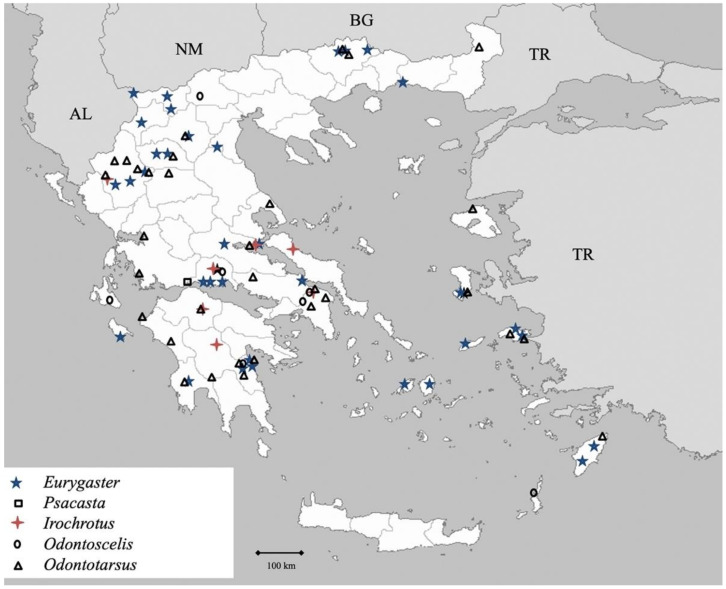
Map showing collecting sites of species of genera *Eurygaster, Psacasta, Irochrotus, Odontoscelis* and *Odontotarsus* during the present study in Greece.

90.*Odontotarsus purpureolineatus* (Rossi 1790)

Achaia [Bouboukas (21.VI.1986)], Aetoloakarnania [Astakos (14.VI.1984), Mytikas-Astakos (23.VI.1986)], Arkadia [Charadros (17.VI.1986)], Attiki [Avlon (7.VI.1978), Marathon (11.VII.1984, 14.V.1985)], Chios [Armolia (20.VI.1987)], Doris [Skaloula (20.VII.1986)], Epirus [Vryssochori (26.V.1981)], Evros [Metaxades (2.VI.1982)], Grevena [Anoixis (31.VII.1984)], Ilia [Kyllini Mt. (21.VII.1982)], Ioannina [Voutsaras (24.VI.1984)], Konitsa [Aoos Riv.-Monastiri (29.V.1981)], Kozani [Vourinos Mt. (18.VIII.1983), (28.VI.1984)], Lesvos [Petra (17.VI.1987)], Messinia [Artemissia (11.VI.1985), Kazarma (4.VII.1984)], Olympos Mt. [Prionia (21.V.1981)], Pilion Mt. [Chania 16.VIII.1980)], Rhodos [Petaloudes) (29.V.1990), Rodopi Mt. [Livaderon (10.VIII.1985), Sidironero (23.V.1983)], Samos [Neochorion (24.VI.1987), Pythagorion (23.VI.1987)], Trikala [Mourgani (19.VI.1985)], Voeotia [Tsoukalades (12.V.1988)], Parnon Mt. (11.VI.1985). Total: 36 specimens.

91.*Odontotarsus robustus* (Jakovlev 1883)

Attiki [Avlon (16.VII.1979), Kifissia (11.VII.1980)], Doris [Skaloula (20.VII.1978, 17.IX.1978, 9.VI.1979, 1.VI.1985)], Pindos [Votonossi (5.IX.1980), Konitsa (22.VI.1986)]. Total: 16 specimens.

92.*Odontotarsus rufescens* (Fieber 1861)

Amphilochia [Anoixiatiko (20.VIII.1985)], Attiki [Avlon (20.V.1987)], Evia [Agios Georgios (21.VI.1980)], Ilia [Krestena (16.V.1985)], Kynouria [Astros (13.VII.1982)], Messinia [Kazarma (4.VII.1984)]. Total: 7 specimens.

**Table 1 insects-13-00749-t001:** Updated checklist of Pentatomoidea of Greece.

Family	Subfamily	Tribe	Species	# in the Present Study	First Reference
Acanthosomatidae	Acanthosomatinae		*Acanthosoma haemorrhoidale haemorrhoidale* (L. 1758)	-	D
			*Cyphostethus tristriatus* (Fabricius 1860)	1	R
			*Elasmostethus interstinctus* (Linnaeus 1758)	-	R
			*Elasmucha grisea grisea* (Linnaeus 1758)	2	D
Cydnidae	Cydninae	Cydnini	*Cydnus aterrimus* (Forster 1771)	3	R
		Geotomini	*Aethus hispidulus* (Klug 1845)	-	D
			*Aethus pilosus* (Herrich-Schaeffer 1834)	-	Lis
			*Byrsinus balcanicus* (Josifov 1986)	-	Josifov a
			*Byrsinus pilosulus* (Klug 1845)	-	D
			*Geotomus brunnipennis* (Wagner 1953)	-	D
			*Geotomus ciliatitylus* (Signoret 1881)	-	D
			*Geotomus elongatus* (Herrich-Schaeffer 1840)	-	R
			*Geotomus punctulatus* (Costa 1847)	-	R
			*Macroscytus brunneus* (Fabricius 1803)	4	R
			*Microporus nigrita* (Fabricius 1794)	-	Lis
	Sehirinae	Amaurocorini	*Amaurocoris curtus* (Brulle 1839)	-	Lis
		Sehirini	*Adomerus maculipes* (Mulsant et Rey 1852)	-	R
			*Canthophorus dubius* (Scopoli 1763)	-	R
			*Canthophorus melanopterus melanopterus* (Herrich-Schaeffer 1835)	-	D
			*Crocistethus basalis* (Fieber 1861)	-	Rieger a
			*Crocistethus waltlianus* (Fieber 1837)	-	Lis
			*Legnotus fumigatus* (A. Costa 1853)	-	D
			*Legnotus limbosus* (Geoffroy 1785)	-	D
			*Legnotus picipes* (Fallen 1807)	-	Günther
			*Ochetostethus balcanicus* (Wagner 1940)	5	D
			*Ochetostethus heissi* (Magnien 2006)	-	Aukema
			*Ochetostethus opacus* (Scholtz 1847)	-	Günther
			*Sehirus luctuosus* (Mulsant et Rey 1866)	-	Lis
			*Sehirus morio* (Linnaeus 1761)	-	D
			*Sehirus ovatus* (Herrich-Schaeffer 1840)	-	D
			*Singeria brevipennis* Wagner 1955	-	D
			*Tritomegas bicolor* (Linnaeus 1758)	6	R
			*Tritomegas sexmaculatus* (Rambur 1839)	7	D
Pentatomidae	Asopinae		*Andrallus spinidens* (Fabricius 1787)	-	Rieger b
			*Arma insperata* (Horváth 1899)	8	D
			*Jalla dumosa* (Linnaeus 1758)	9	D
			*Perillus bioculatus* (Fabricius 1775)	-	Pericart
			***Picromerus bidens**** (Linnaeus 1758)	**10**	**X**
			*Picromerus conformis* (Herrich-Schaeffer 1841)	11	Günther
			*Picromerus nigridens* (Fabricius 1803)	-	R
			*Pinthaeus sanguinipes* (Fabricius 1787)	-	D
			*Troilus luridus* (Fabricius 1775)	12	R
			*Zicrona caerulea* (Linnaeus 1758)	13	R
	Pentatominae	Aeliini	*Aelia acuminata* (Linnaeus 1758)	16	R
			*Aelia albovittata* (Fieber 1868)	17	Rider
			*Aelia furcula* (Fieber 1868)	-	D
			***Aelia germari**** (Küster 1852)	**18**	**X**
			*Aelia klugii* (Hahn 1831)	19	D
			*Aelia rostrata* (Boheman 1852)	20	D
			*Aelia virgata* (Herrich-Schaeffer 1841)	21	D
			*Neottiglossa bifida* (Costa 1847)	47	D
			*Neottiglossa flavomarginata* (Lucas 1849)	48	R
			*Neottiglossa leporina* (Herrich-Schaeffer 1830)	49	R
			***Neottiglossa lineolata**** (Mulsant et Rey 1852)	**50**	**X**
			***Neottiglossa pusilla**** (Gallen 1789)	**51**	**X**
		Cappaeini	*Halyomorpha halys* (Stål 1855)	41	Milonas et Partsinevelos
		Carpocorini	*Antheminia lunuluta* (Goeze 1778)	22	D
			*Antheminia varicornis* (Jakovlev 1874)	-	D
			*Brachynema cinctum* (Fabricius 1775)	24	R
			*Brachynema germarii* (Kolenati 1846)	-	D
			*Carpocoris (Carpocoris) fuscispinus* (Boheman 1851)	25	Rider
			*Carpocoris (Carpocoris) mediterraneus mediterraneus* (Tamanini 1958)	26	D
			*Carpocoris (Carpocoris) pudicus* (Poda 1761)	27	D
			*Carpocoris (Carpocoris) purpureipennis* (De Geer 1773)	-	R
			*Chlorochroa juniperina juniperina* (Linnaeus 1758)	-	D
			*Chlorochroa pinicola* (Mulsant et Rey 1852)	-	Ribes et Pagola-Carte
			*Chroantha ornatula* (Herrich-Schaeffer 1842)	-	D
			*Codophila varia* (Fabricius 1787)	29	R
			*Dolycoris baccarum* (Linnaeus 1758)	30	R
			*Holcogaster fibulata* (Germar 1831)	42	R
			*Holcostethus albipes* (Fabricius 1781	43	D
			*Holcostethus sphacelatus* (Fabricius 1794)	44	Günther
			*Palomena prasina* (Linnaeus 1761)	53	R
			*Peribalus strictus strictus* (Fabricius 1803)	55	R
			*Staria lunata* (Hahn 1835)	66	R
		Eysarcorini	***Eysarcoris aeneus**** (Scopoli 1763)	**39**	**X**
			*Eysarcoris ventralis* (Westwood 1837)	40	R
			*Stagonomus amoenus* (Brulle 1832)	64	R
			*Stagonomus bipunctatus* (Linnaeus 1758)	65	R
			*Stagonomus devius* (Seidenstücker 1965)	-	Rieger b
			*Stagonomus grenieri* (Signoret 1865)	-	Derjanschi et Pericart
			*Stagonomus venustissimus* (Schrank 1776)	-	Rieger c
		Halyini	*Aphodiphus amygdali* (Germar 1817)	23	R
			*Mustha spinosula* (Lefebvre 1831)	45	R
		Mecideini	*Mecidea lindbergi* (Wagner 1954)	-	D
		Menidini	*Neostrachia bisignata* (Walker 1867)	46	D
		Pentatomini	*Acrosternum arabicum* (Wagner 1959)	-	Rider
			*Acrosternum heegeri* (Fieber 1861)	14	D
			*Acrosternum malickyi* (Josifov et Heiss 1989)	-	Josifov et Heiss
			*Acrosternum millierei* (Mulsant et Rey 1866)	15	D
			*Nezara viridula* (Linnaeus 1758)	52	R
			*Pentatoma (Pentatoma) rufipes* (Linnaeus 1758)	54	R
			*Rhaphigaster nebulosa* (Poda 1761)	57	R
		Piezodorini	*Piezodorus lituratus* (Fabricius 1794)	56	R
		Sciocorini	*Dyroderes umbraculatus* (Fabricius 1775)	31	R
			*Sciocoris convexiusculus* (Puton 1874)	-	Hoberlandt
			*Sciocoris cursitans cursitans* (Fabricius 1794)	58	D
			*Sciocoris deltocephalus* (Fieber 1861)	59	D
			*Sciocoris helferii* (Fieber 1851)	60	D
			*Sciocoris macrocephalus* (Fieber 1851)	61	D
			*Sciocoris maculatus* (Fieber 1851)	62	R
			*Sciocoris microphthalmus* (Flor 1860)	-	Derjanschi et Pericart
			*Sciocoris ochraceus* (Fieber 1861)	-	Rider
			*Sciocoris pictus* (Wagner 1959)	-	Derjanschi et Pericart
			*Sciocoris sulcatus* (Fieber 1851)	63	R
		Strachiini	*Bagrada* (*Nitilia*) *abeillei* (Puton, 1881)	-	Derjanschi et Pericart
			*Bagrada* (*Nitilia*) *stolida* (Herrich-Schaeffer 1839)	-	R
			*Eurydema blanda* (Horváth 1903)	-	Heckmann
			*Eurydema eckerleini* (Josifov 1961)	32	Josifov b
			*Eurydema fieberi* (Schummel 1837)	33	D
			*Eurydema oleracea* (Linnaeus 1758)	34	D
			*Eurydema ornata* (Linnaeus 1758)	35	R
			*Eurydema rotundicollis* (Dohrn 1860)	-	Günther
			*Eurydema rugulosa* (Dohrn 1861)	36	R
			*Eurydema spectabilis* (Horváth 1882)	37	D
			*Eurydema ventralis* (Kolenati 1846)	38	D
			*Stenozygum coloratum* (Klug 1845)	-	D
			*Trochiscocoris rotundatus rotundatus* (Horváth 1895)	-	Günther
	Podopinae	Graphosomatini	*Ancyrosoma leucogrammes* (Gmelin 1790)	67	R
			*Derula flavoguttata* (Mulsant et Rey 1856)	68	R
			*Graphosoma italicum italicum* (Müller 1766)	69	R
			*Graphosoma semipunctatum* (Fabricius 1775)	70	R
			*Leprosoma inconspicuum* (Baerensprung 1859)	71	D
			*Tholagmus flavolineatus* (Fabricius 1798)	76	R
			*Ventocoris (Selenodera) achivus* (Horváth 1889)	77	R
			*Ventocoris (Ventocoris) rusticus* (Fabricius 1781)	78	R
			*Vilpianus galii* (Wolf 1802)	-	Pericart
		Podopini	*Podops curvidens* (Costa 1843)	72	D
			***Podops inunctus**** (Fabricius 1775)	**73**	**X**
			*Podops rectidens* (Horváth 1883)	74	R
			*Tarisa flavescens* (Amyot et Serville 1843)	-	D
			***Tarisa pallescens**** (Jakovlev 1871)	**75**	**X**
Plataspididae	Plataspidinae	Plataspidini	*Coptosoma scutellatum* (Geoffroy 1785)	79	D
Scutelleridae	Elvisurinae		*Solenosthedium bilunatum* (Lefebvre 1827)	-	D
	Eurygastrinae	Eurygastrini	*Eurygaster austriaca* (Schrank 1778)	80	D
			*Eurygaster dilaticollis* (Dohrn 1860)	81	D
			***Eurygaster hottentotta**** (Fabricius 1775)	**82**	**X**
			*Eurygaster integriceps* (Puton 1881)	83	R
			*Eurygaster maura* (Linnaeus 1758)	84	R
			*Eurygaster testudinaria testudinaria* (Geoffroy 1785)	85	D
			*Psacasta (Psacasta) exanthematica exanthematica* (Scopoli 1763)	86	R
			*Psacasta (Cryptodontus) tuberculata* (Fabricius 1781)	-	D
	Odontoscelinae		*Irochrotus maculiventris* (Germar 1839)	87	D
			*Odontoscelis (Odontoscelis) byrrhus* (Seidenstücker 1972)	-	Göllner-Scheiding a
			*Odontoscelis (Odontoscelis) fuliginosa* (Linnaeus 1761)	88	R
			*Odontoscelis (Odontoscelis) lineola* (Rambur 1839)	89	D
			*Odontoscelis (Odontoscelis) minuta* (Jakovlev 1882)	-	Göllner-Scheiding a
	Odontotarsinae	Odontotarsini	*Odontotarsus caudatus* (Burmeister 1835)	-	D
			*Odontotarsus crassus* (Kiritschenko 1966)	-	Göllner-Scheiding b
			*Odontotarsus freyi* (Puton 1882)	-	D
			*Odontotarsus grammicus* (Linnaeus 1767)	-	R
			*Odontotarsus parvulus* (Horváth 1917)	-	D
			*Odontotarsus plicatulus* (Horváth 1906)	-	Göllner-Scheiding c
			*Odontotarsus purpureolineatus* (Rossi 1790)	90	D
			*Odontotarsus robustus* (Jakovlev 1884)	91	D
			*Odontotarsus rufescens* (Fieber 1861)	92	D
Thyreocoridae	Thyreocorinae		*Thyreocoris scarabaeoides* (Linnaeus 1758)	-	R

* Bold letters and asterisk indicate the new species record for Greece. D (Drosopoulos 1980), R (Reuter 1891), Lis (Lis 1999), Josifov a (Josifov 1986b), Josifov b (Josifov 1961), Josifov et Heiss (Josifov et Heiss 1989), Rieger a (Rieger 1995), Rieger b (Rieger 2007), Rieger c (Rieger 2012), Günther (Günther 1990), Aukema (Aukema et al. 2013), Pericart (Pericart 2010), Göllner-Scheiding a (Göllner-Scheiding 1986), Göllner-Scheiding b (Göllner-Scheiding 1990), Göllner-Scheiding c (Göllner-Scheiding 2006), Rider (Rider 2006), Milonas et Partsinevelos (Milonas and Partsinevelos 2014), Ribes et Pagola-Carte (Ribes and Pagola-Carte 2013), Derjanschi et Pericart (Derjanschi and Pericart 2005), Hoberlandt (Hoberlandt 1997), Heckmann (Heckmann et al. 2005), X (present study).

## 4. Discussion

In Greece, until the present work, very few studies have been conducted on the Pentatomoidea fauna, in which 150 Pentatomoidea species are referred. Except two overall studies performed by Drosopoulos [43] and Ramsey [44], in 1980 and 2019, respectively, only sporadic references are made by numerous researchers, considering the Greek fauna as part of a wider Balkans. In the present study, 8 species are recorded in Greece for the first time, raising the number of the Greek fauna to 158 species. It has to be mentioned that the new records refer to specimens that were collected during 1982–1983 and placed in the historic collection of Sakis Drosopoulos. Our study provides one step toward enriched knowledge of the Pentatomoidea fauna of Greece, which will hopefully trigger more comprehensive taxonomic study of the entire superfamily.

## Figures and Tables

**Figure 1 insects-13-00749-f001:**
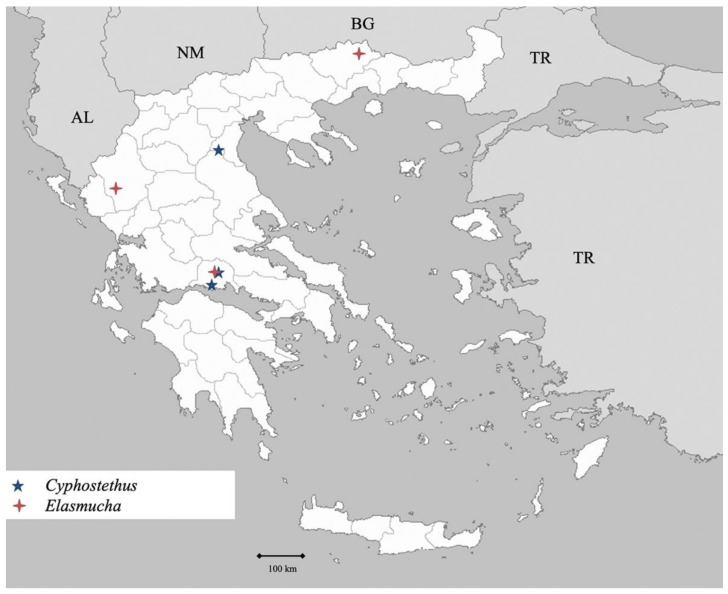
Map showing collecting sites of species of genera *Cyphostethus* and *Elasmucha* during the present study in Greece (AL = Albania, NM = North Macedonia, BG = Bulgaria, TR = Turkey).

**Figure 2 insects-13-00749-f002:**
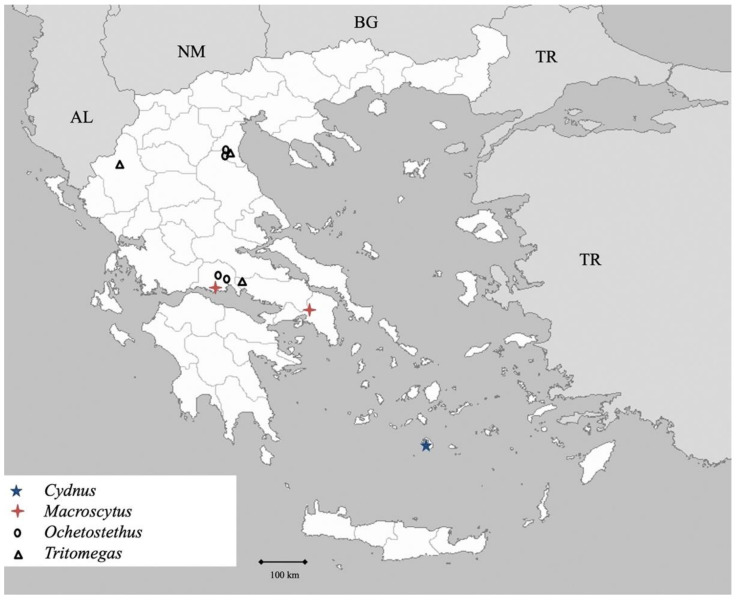
Map showing collecting sites of species of genera *Cydnus, Macroscytus, Ochetostethus* and *Tritomegas* during the present study in Greece.

**Figure 7 insects-13-00749-f007:**
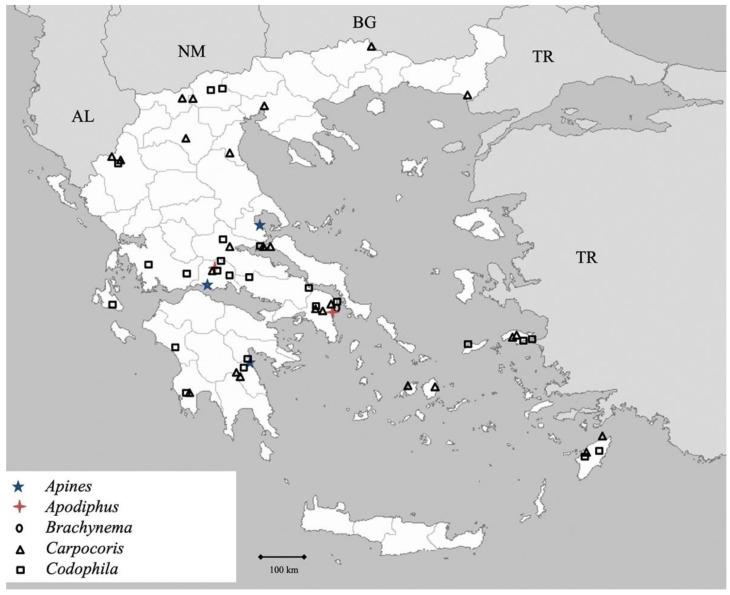
Map showing collecting sites of species of genera *Apines, Apodiphus, Brachynema, Carpocoris* and *Codophila* during the present study in Greece.

**Figure 8 insects-13-00749-f008:**
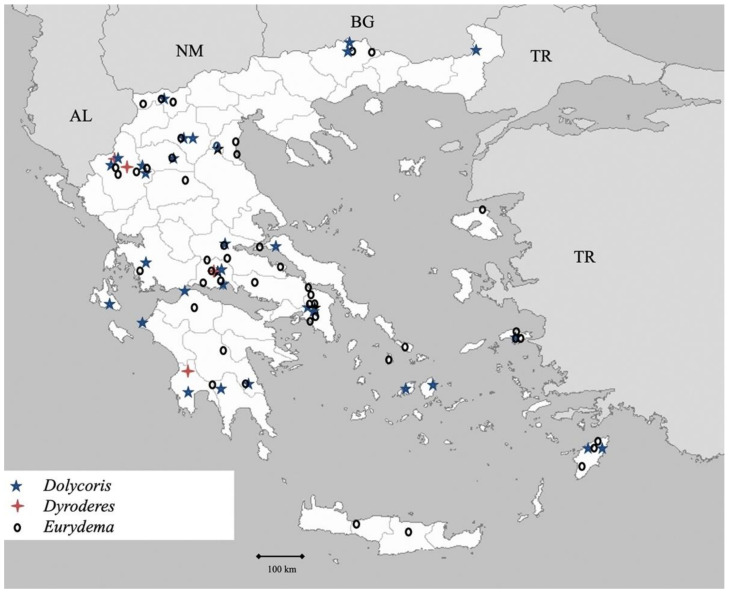
Map showing collecting sites of species of genera *Dolycoris, Dyroderes* and *Eurydema* during the present study in Greece.

**Figure 14 insects-13-00749-f014:**
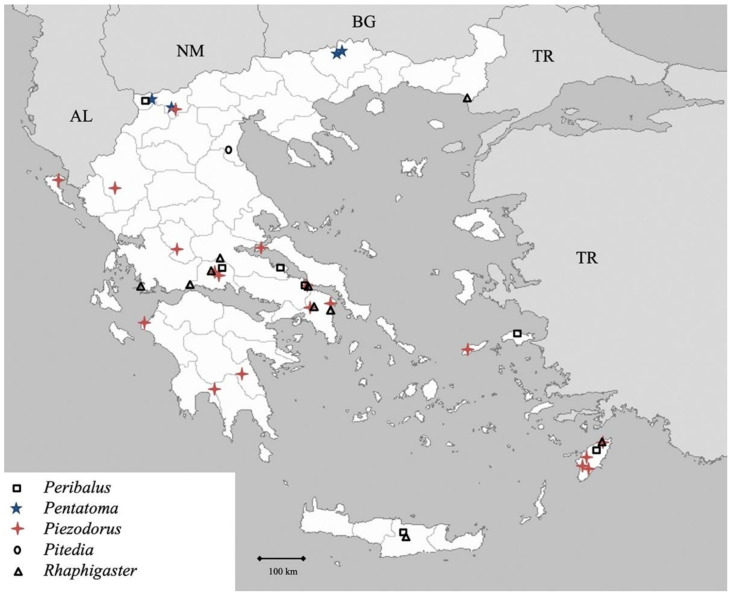
Map showing collecting sites of species of genera *Peribalus, Pentatoma, Piezodorus, Pitedia and Rhaphigaster* during the present study in Greece.

**Figure 15 insects-13-00749-f015:**
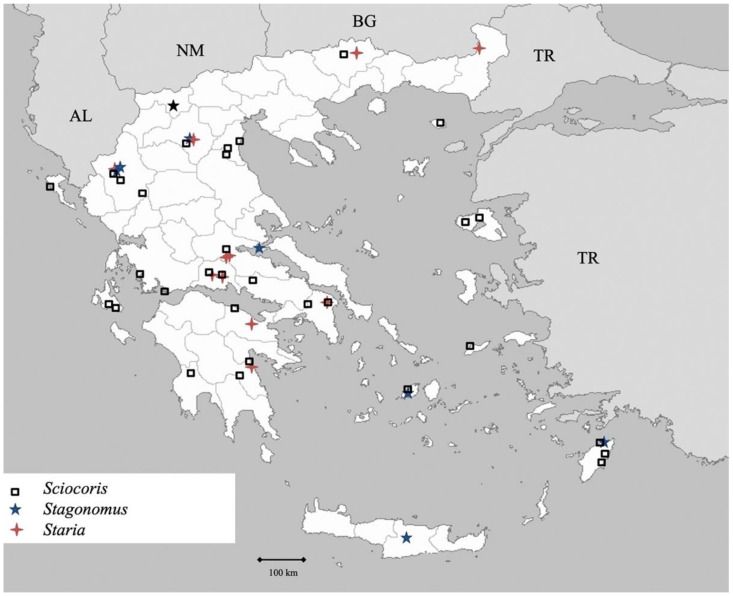
Map showing collecting sites of species of genera *Sciocoris, Stagonomus* and *Staria* during the present study in Greece.

**Figure 16 insects-13-00749-f016:**
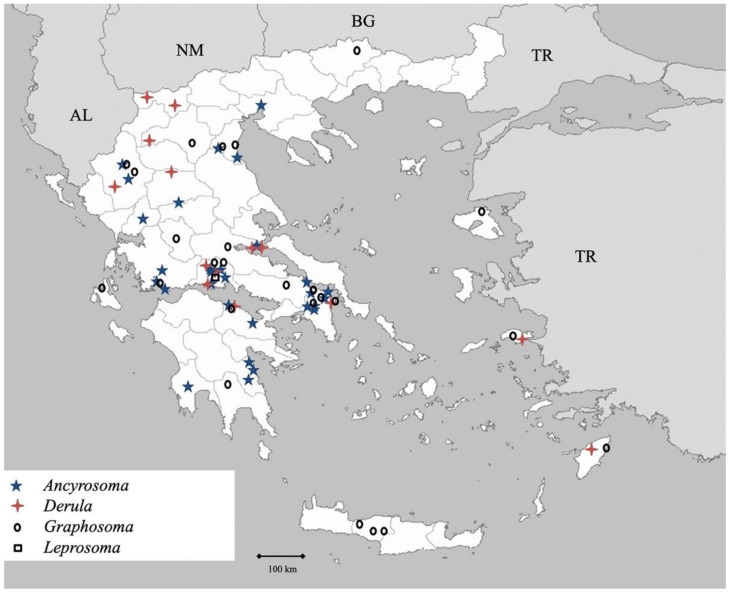
Map showing collecting sites of species of genera *Ancyrosoma, Derula, Graphosoma* and *Leprosoma* during the present study in Greece.

**Figure 20 insects-13-00749-f020:**
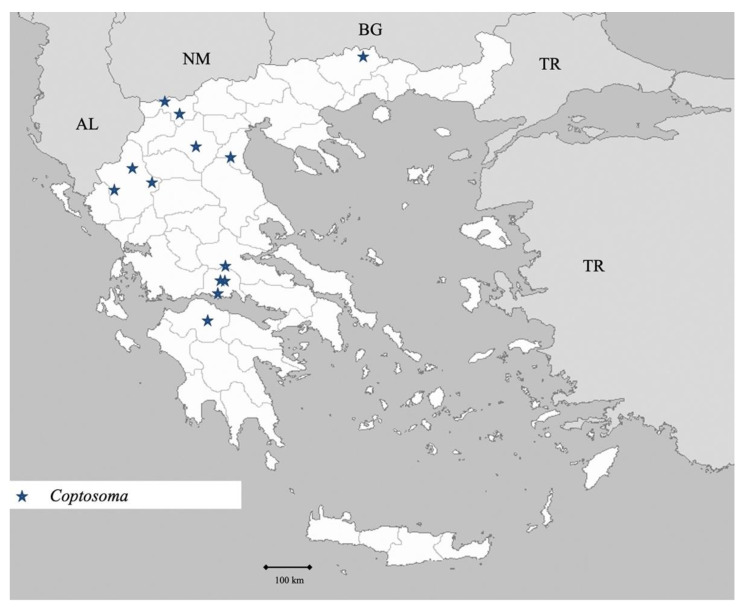
Map showing collecting sites of species of genus *Coptosoma* during the present study in Greece.

## Data Availability

Data is contained within the article.

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
