# Peer review of "New Records and Updated Checklist of the Pentatomoidea (Hemiptera: Heteroptera) of Greece"

_insects, 2022, doi:10.3390/insects13080749_

Round 1

Reviewer 1 Report

see attached

Author Response

Lines 16-23 Please check if there are 6 or 8 new species found. In Line 16 is reported as six species and again in Line 18, yet eight species are listed in the abstract, and there are 8 species in bold in Table 1.

Corrected to 8 species

Line 27 Introduction. It would be nice to have a paragraph or two expanding on the Pentatomidae as an insect family, why they should be studied in Greece, particularly given the presence of the brown marmorated stink bug in Europe, Greece and its importance as a global pest of crops.

I.e. final statement before the aim “Given the worldwide problem of the brown marmorated stink bug pest status (Leskey and Nielson 2018) and its presence in Greece [35], understanding this family in Greece may assist in identifying this pest species from native ones, and provide more targeted response for future detections of this pest”.

  • Leskey TC, Nielsen AL. 2018. Impact of the invasive brown marmorated stink bug in North America and Europe: History, biology, ecology, and management. Annual Review of Entomology 63: 599-618.

Paragraph and literature inserted

Line 33 insert comma “studies,” " Comma inserted

Line 40 remove repetition in the manuscript where necessary “binocular microscopes (Carl Zeiss Stemi 305 and Olympus CX23)”. Sentence rephrased

Line 57 change to “around Vouchorina; Korydallos and Votonossi”, Lines 53-132 The around is used too often making this section hard to read. Please remove from the brackets and elsewhere if possible. Location description changed according to the reviewers comments.

Lines 135-140 “of Pentatomoidea” insert after “5 families”: Words inserted

It is unclear what the letters on the map are referring to in the Figures M1-M14. Please place their meaning at least in Figure M1. Symbols meaning added in Figure M1 (which is renamed to Figure 1 after reviewers suggestion).

Each genus or species is numbered and listed in the result section in that order. However, the numbers are not referred to in the text or the figures. It is suggested that authors place the appropriate numbers next to their corresponding genus species names in the Table 1. This would better link the table to the results and allow readers to match up this information about each genus and species reported on in this study. Numbers of species added in table 1 as suggested

Reviewer 2 Report

I my view the paper makes a significant contribution to our knowledge of the Pentatomoidea of Greece, based on extensive collections. I do not see any value in the list of places where collections were made: this is shown in the maps and given with the species anyway, so I do not think the  duplication is justified and suggest you remove it. I have made a list of suggested improvements, which I attach sperately. I have made no attempt to check the spelling of locality names as the places are unknown to me.

Line 2 Hemiptera (!!) 15 Delete: In the present work 16 Delete: In this 29 Write: families of which the Cynidae ... (otherwise you need a reference after "families" 33 Write: studies, one.... which update 38 Write: sweeping nets and glass containers, or by.... 40 Write: observation under Carl Zeiss 305 and Olympus CX23 binocular microscopes by maceration (no comma). (I do not think the names of the microscopes are necessary) 42 I do not like : according to (39-43). Perhaps write: according to keys and descriptions in references 39-42 44 Use established for created 49 I would prefer: by Drosopoulus et al. (44) 52-133 For me a lost like this is meaningless. Localities should be given with the species.{I have not checked spelling of locality names) 704 Table1: see Byrsinus pilosulus: Josifov Josinov? 731 No date given. There may be others without a date, please check 759-766. All German words should not start with capital letters, only nouns. 775 Not capital letters for all words General: Some dates are in bold, others not. Be consistent

Author Response

Line 2 Hemiptera (!!): Word corrected, apologies!

L15 Delete: In the present work: Sentence deleted

L16 Delete: In this. Words deleted

L29 Write: families of which the Cynidae ... (otherwise you need a reference after "families". Rephrased as suggested

L33 Write: studies, one.... which update. Sentence rephrased

L38 Write: sweeping nets and glass containers, or by.... Sentence rephrased

L40 Write: observation under Carl Zeiss 305 and Olympus CX23 binocular microscopes by maceration (no comma). (I do not think the names of the microscopes are necessary). Sentence rephrased. However, the brands remained for more info, if the reviewer is OK with this.

L42 I do not like : according to (39-43). Perhaps write: according to keys and descriptions in references. We fully understand. Unfortunately it is the style of the citation which creates this awkward sentence. We rephrased as suggested.

L44 Use established for created: Change addressed

L49 I would prefer: by Drosopoulus et al. (44) Changed addressed

L52-133 For me a lost like this is meaningless. Localities should be given with the species.{I have not checked spelling of locality names). Localities description changed after Reviewer 1 and 2 suggestions. We prefer to keep it since it gives details which make the map figures easier to read

L704 Table1: see Byrsinus pilosulus: Josifov Josinov? Author name corrected

L731 No date given. There may be others without a date, please check. Date added, others checked and found to be OK.

L759-766, All German words should not start with capital letters, only nouns. We agree with the reviewer's suggestion, but the instructions of the journal suggest the authors to use the specific style (to capitalize each word), which is applied automatically through Zotero template.

L775 Not capital letters for all words General: Some dates are in bold, others not. Be consistent. Similarly with the previous answer. Date font changed to bold.

Reviewer 3 Report

The authors have drafted a straightforward manuscript detailing the 101 species of Pentatomoidea they identified in Greece. The manuscript is well written, and the language and grammar is clean.

In the simple summary and abstract, I was confused as to why the authors state that only six new species were identified, while eight species were in fact new to the records for Greece.

The introduction is succinct but sufficient for this type of manuscript, and the authors do properly cite the previous published surveys of Pentatomoidea in Greece.

The materials and methods section is succinct as well. Here I think it would be a good to include a short sentence detailing the range of years these samples come from.

The figures showing the locality of collected specimens are a nice addition. The authors could choose to shrink their symbol sizes to more clearly see each shape. This is not necessary though as most shapes can be identified and remaking the maps could take significant time. The figures depicting new species are clear except for Figure 1 (Picromerus bidens). That photo could be retaken so it matches the quality of the other seven photos.

The results and discussion is clear and sufficient for a species checklist.

Overall, this manuscript should be considered for publication after minor revisions. I list the suggested revisions below.

Line 16: Eight species are new records for Greece per line 139, but here in the simple summary it states only six new species were identified.

Line 18: Same as above.

Line 22: Will you in fact include drawings of these insect species? I did not see them in this current submission.

Lien 38-46: I suggest adding the range of years these insects were collected in.

Line 49: I suggest rewording this line. Perhaps “Collection sites for each species are presented in the same fashion as [44].”

Line 194: This image of Picromerus bidens is blurry. I suggest retaking the photo for this specimen.

Map Figures: Where many collections were made in nearby locations, the overlap of symbols indicating genera makes it difficult to see what shape each symbol is. Those figures could be made easier to read if the size of each symbol was decreased. This is not a necessary change, however.

Author Response

Lines 16, 18: Eight species are new records for Greece per line 139, but here in the simple summary it states only six new species were identified. Number of new species corrected

Line 22: Will you in fact include drawings of these insect species? I did not see them in this current submission. Word drawings removed

Line 38-46: I suggest adding the range of years these insects were collected in. Range of collection period added.

Line 49: I suggest rewording this line. Perhaps “Collection sites for each species are presented in the same fashion as [44].” Sentence rephrased according to all the Reviewers comments.

Line 194: This image of Picromerus bidens is blurry. I suggest retaking the photo for this specimen. Picture retaken and replaced in the manuscript

Map Figures: Where many collections were made in nearby locations, the overlap of symbols indicating genera makes it difficult to see what shape each symbol is. Those figures could be made easier to read if the size of each symbol was decreased. This is not a necessary change, however. We fully understand the reviewer. Thus, we changed most of the symbols within the Figures in order to be less bold and more readable (two symbols were switched to color).